

# Effects of including the adjoint sea ice rheology on estimating Arctic ocean-sea ice state

Guokun Lyu[1], Armin Koehl[2], Xinrong Wu[3], Meng Zhou[1,4], and Detlef Stammer[2]
[1]Shanghai Key Laboratory of Polar Life and Environment Sciences, School of Oceanography, Shanghai Jiao Tong
University, Shanghai, China
[2]Center for Earth System Research and Sustainability (CEN), University of Hamburg, Hamburg, Germany
[3]Key Laboratory of Marine Environmental Information Technology, National Marine Data and Information Service,
Tianjin, China
[4]MNR Key Laboratory for Polar Science, Polar Research Institute of China, Shanghai, China
*Correspondence to*: Guokun Lyu (guokun.lyu@sjtu.edu.cn)
**Abstract.** The adjoint assimilation method has been applied to coupled ocean and sea ice models for sensitivity studies
and Arctic state estimations. However, the accuracy of the adjoint model is degraded by simplifications of the adjoint
of the sea ice model, especially the adjoint sea ice rheologies. As part of ongoing developments in coupled ocean and
sea ice estimation systems, we incorporate and approximate the adjoint of viscous-plastic sea ice dynamics (adjoint-
VP) and compare it with the adjoint of free drift sea ice dynamics (adjoint-FD) through assimilation experiments.
Using the adjoint-VP results in a further cost reduction of 7.9% in comparison to adjoint-FD, with noticeable
improvements in the ocean temperature over the open water and the intermediate layers of the Arctic Ocean. Adjoint-
VP more efficiently adjusts the uncertain model inputs more efficiently than does adjoint-FD by involving different
sea ice retreat processes. For instance, adjoint-FD melts the sea ice up to 1.0 m in the marginal seas from May to June
by over-adjusting air temperature (>8 ℃); adjoint-VP reproduces the sea ice retreat with smaller adjustments to the
atmospheric state within their prior uncertainty range. These developments of the adjoint model here lay the foundation
for further improving Arctic Ocean and sea ice estimationS by comprehensively adjusting the initial conditions,
atmospheric forcings, and uncertain parameters of the model.

## 1 Introduction

The Arctic Ocean has experienced drastic changes, including rapidly declining sea ice (Comiso et al., 2008; Kwok, 2018), increased inventory of freshwater in the western Arctic (Proshutinsky et al., 2019), enhanced warm inflows from the Pacific Ocean (Woodgate et al., 2012) and the Atlantic Ocean (Polyakov et al., 2017; Quadfasel et al., 1991), and increased ocean primary productivity (AMAP, 2021), and has been migrating to a new state over the past decades.These changes potentially impact the climate and weather of the Northern Hemisphere (Ma et al., 2022; Overland et al., 2021).

In recent years, progress has been made in satellite techniques (e.g., Kaleschke et al., 2001; Spreen et al., 2008), in-situ observations (e.g., Toole et al., 2016; Morison et al., 2007; Polyakov et al., 2017; Proshutinsky et al., 2009; Schauer et al., 2008), and coupled ocean and sea ice models. However, the lack of extensive Arctic observations, especially direct observations of the state variables and fluxes through the water column, and the deficiencies in the coupled ocean and sea ice model still obscure our understanding of the Arctic sea ice changes and extremes. Accurate predictions of sea ice are therefore limited (e.g., Yang et al., 2020).

To fill the gaps, research groups have applied data assimilation techniques to ingest available observations into coupled ocean and sea ice models. The resulting reanalyses are assumed to have higher accuracy since as the development of models and data assimilation methods progress and the observation numbers increase. Most of Arctic coupled ocean and sea ice data assimilation and operational forecasting systems use statistical methods such as optimal interpolation (e.g., Lindsay and Zhang, 2006) and ensemble Kalman filters (e.g., Mu et al., 2018; Sakov et al., 2012). The advantage of these statistical methods is that they ensure a local fit to available observations (within prior uncertainties of both model and observations). However, away from the observations and for the unobserved variables, these methods rely on the inaccurate spatial covariance of model states for interpolation. In addition, these algorithms can introduce artificial sink/source terms to the numerical models.

Over recent decades, an adjoint method with a large assimilation window (years to decades) has been developed in the framework of Estimating the Circulation and Climate of the Ocean (ECCO, Heimbach et al., 2019; Stammer et al., 2002; Wunsch and Heimbach, 2007) to create dynamically consistent ocean reanalyses. This method iteratively minimises a cost function that measures the model-data "distance" by adjusting model uncertain inputs (control variables). The use of an adjoint model (adjoint of the tangent linear approximation of the nonlinear model) as a spatiotemporal interpolator distinguishes this method from the statistical methods. The resulting reanalysis completely follows the model governing equations without having to consider artificial source/sink terms. However, the qualities of the reanalysis datasets depend on the accuracy of the tangent linear approximation.

Despite the application of the coupled ocean and sea ice adjoint model in sensitivity studies (Heimbach et al., 2010; Kauker et al., 2009; Koldunov et al., 2013) and reanalyses (Fenty and Heimbach, 2013; Koldunov et al., 2017; Lyu et al., 2021b; Nguyen et al., 2021), we have to omit the adjoint of sea ice dynamics (Fenty et al., 2017; Nguyen et al., 2021) or simplify it to the adjoint of a free-drift sea ice model (Koldunov et al., 2017; Lyu et al., 2021b) to ensure numerical stability of the adjoint model. Toyoda et al. (2019) noted that further including the adjoint of sea ice rheology results in a much weaker evolution of sensitivity to sea ice velocity by O ($10^2$) in the central Arctic Ocean

than the adjoint of free-drift sea ice dynamics. It is expected that including the adjoint of sea ice rheology could better project the model-data misfits to the control variables and potentially improve the quality of the reanalysis.

In this study, we incorporate and stabilise the adjoint of viscous-plastic sea ice dynamics (Hibler, 1979; Zhang and Hibler 1997), building on prior developments of the coupled ocean and sea ice model and assimilation system (Koldunov et al., 2017; Lyu et al., 2021b). Using the unprecedented sea ice retreat process in 2012 as an example, we evaluate the impacts of using the approximated adjoint of viscous-plastic sea ice dynamics on estimating the Arctic ocean, sea ice, and sea-ice retreat processes.

The paper is organised as follows. In Section 2, we introduce the model configurations and assimilation experiments. We assess the assimilation results in terms of the residual errors in Section 3. We examine adjustments of the control variables in Section 4 and compare the sea ice retreat process in the assimilation runs in Section 5. Section 6 summarises the results of this study and discusses the potential for a further development of global and Arctic state and parameter estimation systems.

## 2 Model Configuration and Experiment Setups

### 2.1 The Coupled Ocean and Sea Ice Modelling and Assimilation System

The data assimilation system is based on the adjoint method in the ECCO framework, using the Massachusetts Institute of Technology ocean general circulation model (MITgcm, Marshall et al., 1997) coupled with the zero-layer dynamic-thermodynamic sea ice model of Hibler (1979). The sea ice dynamics are based on a viscous-plastic sea ice rheology and are solved using a line successive over-relaxation algorithm (Zhang and Rothrock, 2000). The thermodynamic sea ice model includes a prescribed sub-grid ice thickness distribution with 7 thickness categories. On top of the ice, a diagnostic snow model is applied which modifies the heat flux and surface albedo, as in Zhang and Rothrock (2000). The thermodynic-dynamic sea ice model simulates changes in sea ice drift (SID), sea ice concentration (SIC), and mean sea ice thickness (in volume per unit area, mean SIT hereinafter). Losch et al. (2010) reformulated the sea ice model on an Arakawa C grid to match the MITgcm oceanic grid and modified the model codes to permit efficient and accurate automatic differentiation. The adjoint of the coupled ocean and sea ice model is generated by the Transformation of Algorithms in FORTRAN (TAF, Giering and Kaminski, 1998).

The pan-Arctic model covers the Arctic Ocean north of the Bering Strait and the Atlantic Ocean at 44° N (enclosed by black lines in Figure 1). In the horizontal direction, we use a curvilinear grid with a resolution of 12~15 km in the Arctic Ocean and ~18 km in the North Atlantic Ocean. In the vertical direction, the system has 50 z-levels ranging from 10 m at the surface to 456 m in the deep ocean. The open boundaries are provided by a 16 km Atlantic-Arctic Ocean simulation (Serra et al., 2010). At the ocean surface, we use the atmosphere state from the National Centers for Environmental Prediction reanalysis 1 (NCEP-RA1, Kalnay et al., 1996) and bulk formulae (Large and Yeager, 2004) to compute the momentum, heat, and freshwater fluxes. A virtual salt flux parameterisation simulates the dilution and salinification of rainfall, evaporation, and river runoff. River runoff is applied near the river mouth with seasonally varying discharge (Fekete et al., 2002). In addition, unresolved vertical mixing is parameterised using the K-Profile scheme of Large et al. (1994). The background coefficients of vertical diffusion and viscosity are set to

$10^{-5} m^2 s^{-1}$ and $5.6 \times 10^{-4}$ $m^2 s^{-1}$, respectively. Biharmonic viscosity with a coefficient of $2.2 \times 10^{11}$ $m^4 s^{-1}$ represents
unresolved sub-grid eddy mixing. The bottom topography is derived from ETOPO2 (Smith and Sandwell, 1997).

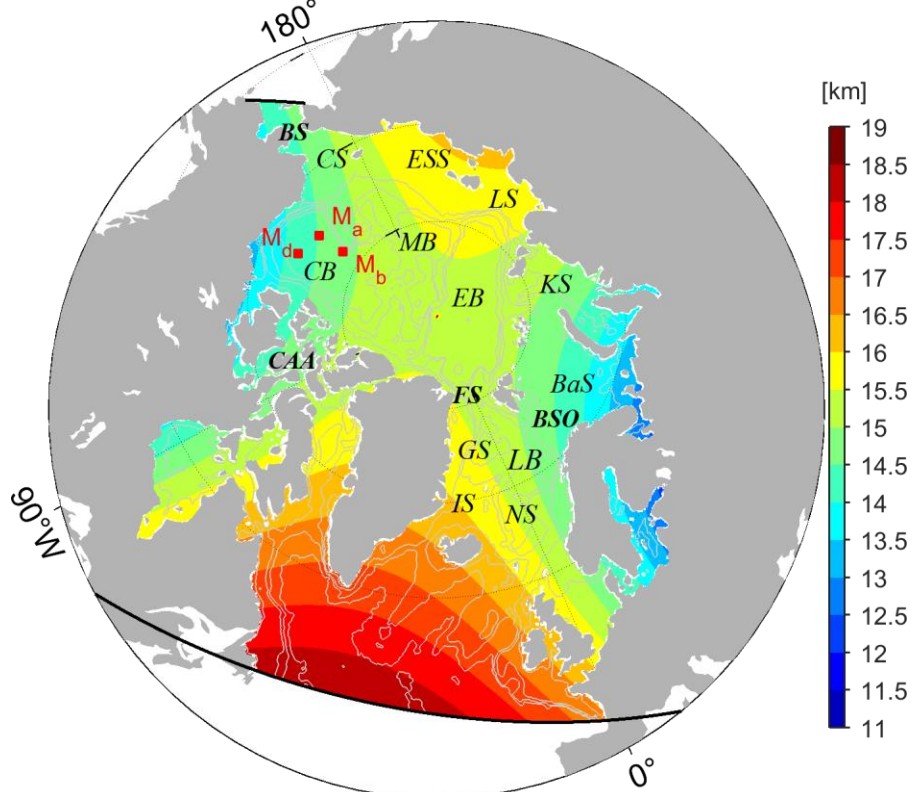

Figure 1. Map of the pan-Arctic regions showing the model domain (enclosed by the black lines) and horizontal
resolutions (shading). The red rectangles show the three moorings ($M_a$, $M_b$, $M_d$) from the Beaufort Gyre Exploration
Project (BGEP). Major basins and straits are labelled as follows: Canada Basin (*CB*), Makarov Basin (*MB*),
Eurasian Basin (*EB*), Chukchi Sea (*CS*), East Siberian Sea (*ESS*), Laptev Sea (*LS*), Kara Sea (*KS*), Barents Sea (*BaS*), Greenland
Sea (*GS*), Lofoten Basin (*LB*), Iceland Sea (*IS*), Norwegian Sea (*NS*), Bering Strait (*BS*), Fram Strait (*FS*), Barents
Sea Opening (*BSO*), and Canadian Arctic Archipelago (*CAA*).
The adjoint method brings the model simulation close to available observations by iteratively adjusting control
variables to minimise a quadric target function *J* (cost function hereafter):
$J(C_{ini}, C_{atm}(t)) = \sum_{t=1}^{T1}[y(t) - E(t)x(t)]^T R^{-2} [y(t) - E(t)x(t)] + C_{ini}^T P^{-2} C_{ini} + \sum_{t=0}^{T1} C_{atm}(t)^T Q_a^{-2} C_{atm}(t)$ (1).
On the right hand side of Equation (1), the first term measures the model-data misfits weighted by the inverse
error covariance matrices ($R^{-2}$). The following section will introduce the available measurements and their
uncertainties (*R*). *y(t)* and *x(t)* are observations and the model state at time *t*, respectively. *E(t)* maps the model state
*x(t)* to the corresponding observations *y(t)*. The last two terms are background terms of the initial condition ($C_{ini}$) and
the time-varying atmospheric forcing ($C_{atm}(t)$) weighted by their inverse error covariance matrices ($P^{-2}$ and $Q_a^{-2}$,
respectively), which penalise their adjustments and provide complete information on the controls. Following Lyu et
al. (2021b), prior uncertainties of the time-varying atmosphere state ($Q_a$) depend on geographic locations. They are
computed as the variance of the nonseasonal variability of the corresponding variables using NCEP-RA1.
For simplicity and the robust performance of this coupled data assimilation system, we choose the initial
conditions ($C_{ini}$), including temperature, salinity, mean SIT, SIC, and daily atmosphere state on the model grid ($C_{atm}(t)$),
which includes 10-m wind vectors, 2-m air temperature, 2-m specific humanity, precipitation, downwelling longwave,
and net shortwave radiation, as the control variables. In the future development of ocean and sea ice state estimation
systems, we further include the river runoff, the open boundary conditions, and model uncertain parameters as control
variables as in previous studies (e.g., Fenty and Heimbach, 2013; Liu et al., 2012). In this study, we use a one-year
assimilation window covering the year 2012, resulting in a total number of $\sim 2.7 \times 10^8$ control variables.
During the optimisation process, the adjoint of the coupled ocean and sea ice model is used to compute the
gradients of the cost function $J$ to the control variables, and a quasi-Newton algorithm (Gilbert and Lemaréchal, 2006)
is used to iteratively reduce the cost function $J$. The optimisation process continues until the cost function cannot be
further reduced.
**2.2 Observations and Prior Uncertainties**
Both satellite and in situ measurements (Table 1) are used to constrain the model simulations. In addition, sea ice
draft measurements by up-looking sonar from the Beaufort Gyre Exploration Project (BGEP, see Figure 1 for the
locations) are used to independently validate the assimilation results.
Prior uncertainties are detailed in our previous Arctic synthesis study (Lyu et al., 2021b). Uncertainties in
temperature and salinity depend on the depth and are set to 0.6 ℃ and 0.3 PSU at the surface and 0.02 ℃ and 0.02
PSU in the deep ocean; SIC uncertainties consist of representation errors (15% within 50 km from the coastlines and
10% over the open water) and instrument errors. Because of higher errors in low SIC and lower errors over open water,
we modify the representation uncertainties by multiplicative factors of 0.85, 1.20, 1.10, and 1.00 for the observed SIC
ranges of 0.00, <15%, 15%–25%, and >25%, respectively.
SIT errors are provided by the datasets and interpolated to our model grid. Sea level anomaly (SLA) uncertainties
are set to 3.0 cm. SID uncertainties are dominated by representation errors and are set to 0.04 m/s. Sea surface
temperature (SST) uncertainties are provided by the datasets. In addition, we reduce the weight of the temperature and
salinity climatology (WOA18) cost components by factors of 20.0 and 10.0, respectively, to avoid overfitting to the
climatology.
**Table 1.** Assimilated measurements.

| Date sets | Resolution | Number | Source |
|---|---|---|---|
| Sea level anomaly | 7.0 km | $7.6 \times 10^5$ | Copernicus Marine Environment Monitoring Service, http://marine.copernicus.eu |
| Sea surface temperature | 25.0 km | $2.0 \times 10^7$ | Remote Sensing System, http://www.remss.com/measurements/sea-surface-temperature/ |
| T&S profiles | – | $5.0 \times 10^5$ | Good et al. (2013), https://www.metoffice.gov.uk/hadobs/en4/ |
| Sea ice concentration | 25.0 km | $3.6 \times 10^7$ | Kaleschke et al. (2001) and Spreen et al. (2008), SSMI (2011-2012), http://icdc.cen.uni-hamburg.de/1/daten/cryosphere.html |
| Sea ice thickness | 25.0 km | $8.9 \times 10^6$ | Ricker et al. (2017), https://spaces.awi.de/pages/viewpage.action?pageId=291898639 |
| Sea ice drift | 62.5 km | $5.8 \times 10^5$ | Lavergne et al. (2019), https://osi-saf.eumetsat.int/products/osi-405-c |
| WOA18 | 1.0° | $2.9 \times 10^7$ | Zweng et al. (2018), https://www.nodc.noaa.gov/OC5/woa18/woa18data.html |

**2.3 Viscos-Plastic Sea Ice Dynamics and Its Adjoint**
In the coupled ocean and sea ice model, the following equation governs sea ice drift:
$$m\frac{d\vec{u}}{dt} = -mf\vec{k} \times \vec{u} + \tau_{air} + \tau_{ocn} - \nabla\emptyset(0) + \nabla \cdot \sigma \tag{2}$$

where $m$ is the sea ice mass and $\vec{u}$ is the sea ice motion vector; $\tau_{air}$ and $\tau_{ocn}$ are the wind and ocean drags,
respectively; $-\nabla\emptyset(0)$ is the tilt of the sea surface; and $\nabla \cdot \sigma$ is the divergence of the ice stress tensor $\sigma_{ij}$(i=1,2),
representing the internal forces of sea ice.
In the viscous-plastic rheology of Hibler (1979), the stress tensor $\sigma_{ij}$ is related to the sea ice strain rate ($\epsilon_{ij}$) and
strength ($P$):
$$\sigma_{ij} = 2\eta(\epsilon_{ij}, P)\epsilon_{ij} + [\zeta(\epsilon_{ij}, P) - \eta(\epsilon_{ij}, P)]\epsilon_{kk}\delta_{ij} - \frac{P}{2}\delta_{ij} \tag{3}$$

where $\delta_{ij}$ is the Kronecker delta ($\delta_{ij} = 1$ if i=j, otherwise 0). $\eta$ and $\zeta$ are the bulk and shear viscosities, expressed
as:
$$\zeta = \frac{P}{2\Delta_{reg}} \tag{4}$$

$$\eta = \frac{P}{2e^2\Delta_{reg}} \tag{5}$$

where
$$\Delta = [(\epsilon_{11}^2 + \epsilon_{22}^2)(1 + e^{-2}) + 2(1 - e^{-2})\epsilon_{11}\epsilon_{22} + 4e^{-2}\epsilon_{12}^2]^{\frac{1}{2}} \tag{6}$$

$e$ is the ratio of normal stress to shear stress and is set to 2.0; $\Delta_{reg} = \max(\Delta, \Delta_{min})$ with $\Delta_{min}$ equals $1.0 \times 10^{-10}$. The
sea ice strain rate is computed as:
$$\epsilon_{ij} = \frac{1}{2}(\frac{\partial u_i}{\partial x_j} + \frac{\partial u_j}{\partial x_i}) \tag{7}.$$

The sea ice strength $P$ depends on mean SIT ($H$) and SIC ($C$):
$$P = P^* H \cdot \exp(-C^* \cdot (1 - C)) \tag{8}$$

$P^*$ and $C^*$ are the ice compressive strength constant and ice strength decay constant and are set to $2.75 \times 10^4$ N m$^{-2}$
and 20.0, respectively.
The dependence of the internal force term ($\nabla \cdot \sigma$) on ice velocity is strongly nonlinear, leading to an unstable
adjoint of the coupled ocean and sea ice system. Therefore, previous studies (Koldunov et al., 2017; Lyu et al., 2021b)
used an adjoint of a free drift sea ice model (without an adjoint of $\nabla \cdot \sigma$). Toyoda et al. (2019) pointed out that the full
adjoint of Equation (2) can be stabilised by eliminating the dependence of bulk and shear viscosities on the strain rate
($\epsilon_{ij}$).
Following the study of Toyoda et al. (2019), we eliminate the dependence of bulk and shear viscosities on $\epsilon_{ij}$ in
the adjoint of Equation (2). In addition, we note that there are still strong sensitivities that hamper the convergence of
optimization. We set the adjoint sensitivities of ice velocity to zero if the local sensitivity is 50 times larger than the
global mean of their absolute values. In addition, we also modify the adjoint model in the following ways to ensure
the stability of the adjoint model over a one-year assimilation window:
1) Disable the K-profile mixing parameterisation scheme;
2) Increase the Laplacian horizontal diffusivity of heat and salinity to 500 m$^2$ s$^{-1}$ and lateral eddy viscosity to
10,000 m$^2$ s$^{-1}$;
3) Apply a spatial filter to sensitivity variables calculated in the adjoint of the thermodynamic sea ice model
(see APPENDIX in Lyu et al. (2021b) for details).
Since the sea ice dynamic model is solved using an iterative line successive over-relaxation solver, we note that
the approximated adjoint of the viscous-plastic sea ice dynamics (adjoint-VP) requires ~1.2 times the computational
cost of using the adjoint of a free-drift sea ice model (adjoint-FD).

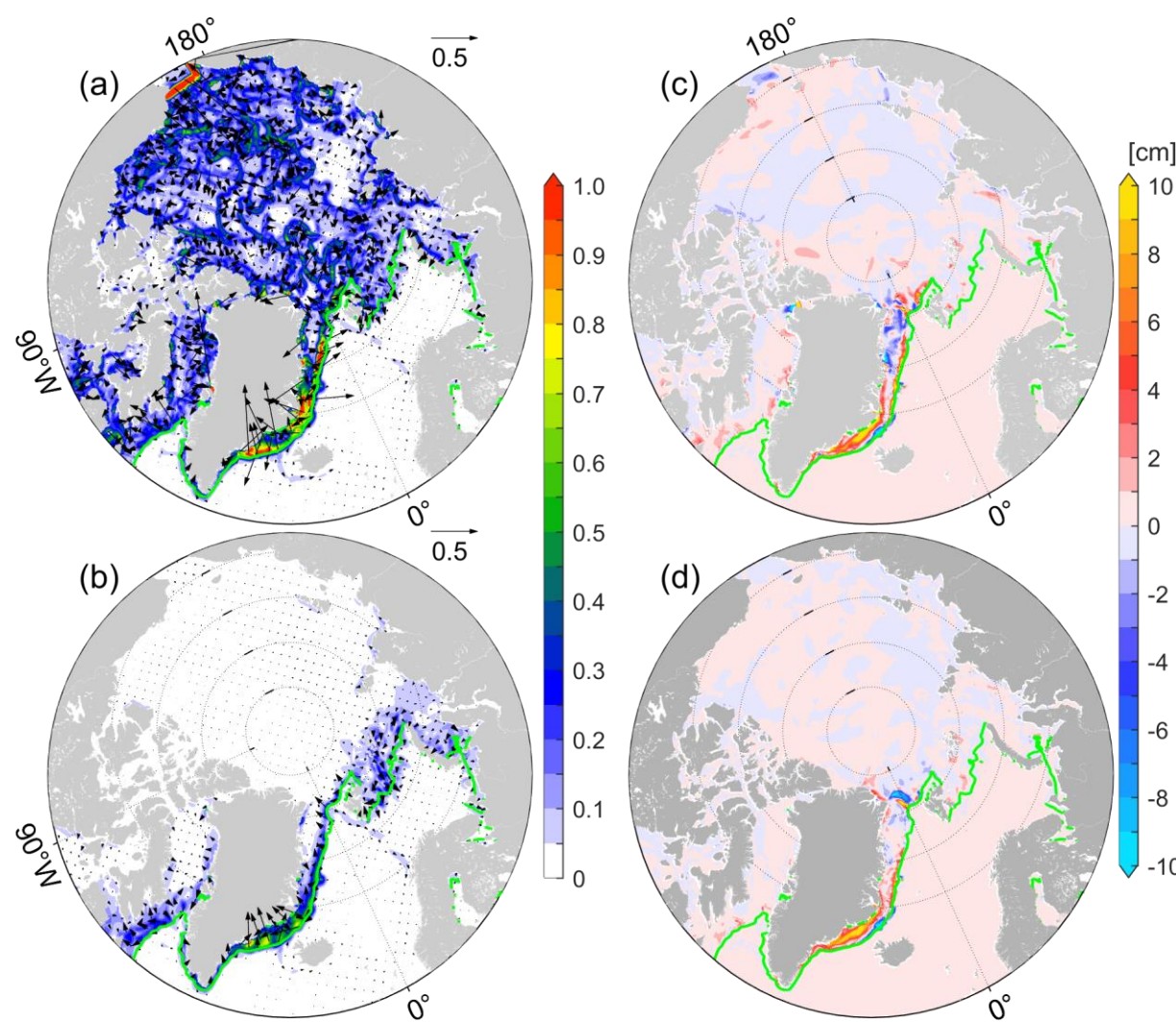


Figure 2. Sensitivities of total sea ice volume to wind vectors (in $0.1 \times km^3$ (m s$^{-1}$)$^{-1}$, shadings represents amplitudes)
using the adjoint of (a) free-drift sea ice dynamics and (b) viscous-plastic sea ice dynamics with modifications in
Section 2.3. Panels (c)-(d) show the mean SIT changes by perturbing the wind with the corresponding adjoint
sensitivities multiplied by a factor of $10^{-8}$. The green lines are the sea ice extents (SIEs, 15% SIC) in January 2012.

Based on adjoint-FD and adjoint-VP, we compute the sensitivities of domain-integrated sea ice volume with
respect to the atmospheric forcings and the initial conditions over the period of January 1 to January 31, 2012. As
reported by Toyoda et al. (2019), adjoint-FD shows much stronger sensitivities to wind than does adjoint-VP (Figure
2a, b) in the central Arctic Ocean. Along the sea ice marginals (SIMs) of the Atlantic sectors, adjoint-VP reveals that
the towards-ice wind anomalies increase total sea ice (Figure 2b) since they prevent ice from drifting to the warm
Atlantic water. However, adjoint-FD shows strong sensitivities along the SIMs of the Atlantic sectors, but both
towards-ice and off-ice wind anomalies appear (Figure 2a), potentially resulting in ice convergence.

202        Furthermore, we add daily wind perturbations, computed by scaling the adjoint sensitivities (Figure 2a, b) so that
the maximum perturbations are 1.0 m s$^{-1}$, to the 6-hourly wind from NCEP-RA1 and examine their impacts on mean
SIT changes. As expected, mean SIT changes are mainly along the SIMs in the Atlantic sectors (Figure 2c, d), and
wind perturbations from adjoint-FD reduces mean SIT northeast of Greenland (Figure 2c). In the central Arctic Ocean
with compact ice, the internal forces $\nabla \cdot \sigma$ oppose the impacts of wind perturbations. Therefore, despite the strong
adjoint sensitivities to the wind in adjoint-FD, we note that the resulting wind perturbations only slightly change the
mean SIT (Figure 2c), which is comparable to that in adjoint-VP (Figure 2d).

209        In addition to overestimating the sensitivities to wind, adjoint-FD may degrade the usefulness of the adjoint
sensitivities in optimisation. Therefore, we perform two assimilation experiments to comprehensively evaluate the
impacts of including the approximate adjoint of sea ice rheology on ocean and sea ice estimations.
**3 Model-Data Misfit Reductions and Residuals**
**3.1 Evaluation of the Optimisation**

214        In this study, we consider iteration 0 the control run. In adjoint-FD and adjoint-VP, the optimisations stall at
iterations 13 and 32, and the further cost function reductions at the last two successive iterations are 0.7% and 0.2%
of the total cost, respectively. After the optimisations, the total cost and norms of the gradients are reduced by 32.3%
and 59.2% , respectively, in adjoint-FD and by 40.2% and 89.3%, respectively, in adjoint-VP.
**Table 2**. Normalised costs and reductions in the two optimisation runs.

| *Cost constituent* | **Control run** | **Adjoint-FD** | | **Adjoint_VP** | |
|---|---|---|---|---|---|
| | Normalised cost (%) | Normalised cost (%) | Percentage reduction (%) | Normalised cost (%) | Percentage reduction (%) |
| $J_{Total}$ | 100 | 67.7 | 32.3 | 59.8 | 40.2 |
| $J_{SLA}$ | 2.2 | 2.1 | 4.6 | 2.1 | 4.6 |
| $J_{SST}$ | 25.3 | 15.4 | 39.1 | 12.9 | 49.0 |
| $J_{profile\_T}$ | 6.9 | 6.5 | 5.8 | 4.3 | 37.7 |
| $J_{profile\_S}$ | 5.8 | 5.9 | -1.7 | 4.5 | 22.4 |
| $J_{SIC}$ | 39.7 | 18.4 | 53.7 | 18.1 | 54.4 |
| $J_{SIT}$ | 3.6 | 3.1 | 13.9 | 2.7 | 25.0 |
| $J_{SID}$ | 4.5 | 4.4 | 2.2 | 4.3 | 4.4 |
| $J_{WOA\_T}$ | 6.6 | 6.6 | 0.0 | 6.2 | 6.1 |
| $J_{WOA\_S}$ | 5.4 | 5.3 | 1.9 | 4.7 | 13.0 |


220        Of the individual cost constituents (Table 2), satellite-observed SST ($J_{SST}$) and SIC ($J_{SIC}$) contribute ~25.3% and
39.7% of the total cost, respectively, which are significantly reduced after optimisation. The costs of the temperature
($J_{profile\_T}$) and salinity ($J_{profile\_S}$) profiles are also considerably reduced, especially in the adjoint-VP. The rest of the cost
constituents are also slightly reduced. Overall, including the adjoint of sea ice rheology further reduces the total cost
by 7.9% and the individual cost constituents, especially $J_{SST}$, $J_{profile\_T}$ and $J_{profile\_S}$. Based on iterations 0, and 13 in
adjoint-FD, and 32 in adjoint-VP of the optimisation, we will focus on the sea ice state and ocean temperature to
evaluate the impacts of using this approximate adjoint of sea ice rheology.

**3.2 Sea Ice State**
**3.2.1 Residual Errors of SIC and SIT**
Satellite visible, infrared, and microwave technologies have been applied to monitor SIC with high frequencies
and quality, which is of high priority in global and Arctic-focused synthesis (Chevallier et al., 2017; Uotila et al.,
2019). Previous studies (Fenty and Heimbach, 2013; Lyu et al., 2021a; Lyu et al., 2021b) indicated that SIC could be
significantly improved by slightly adjusting the atmospheric forcings. Here, we explore the residual errors in the
optimisation runs.

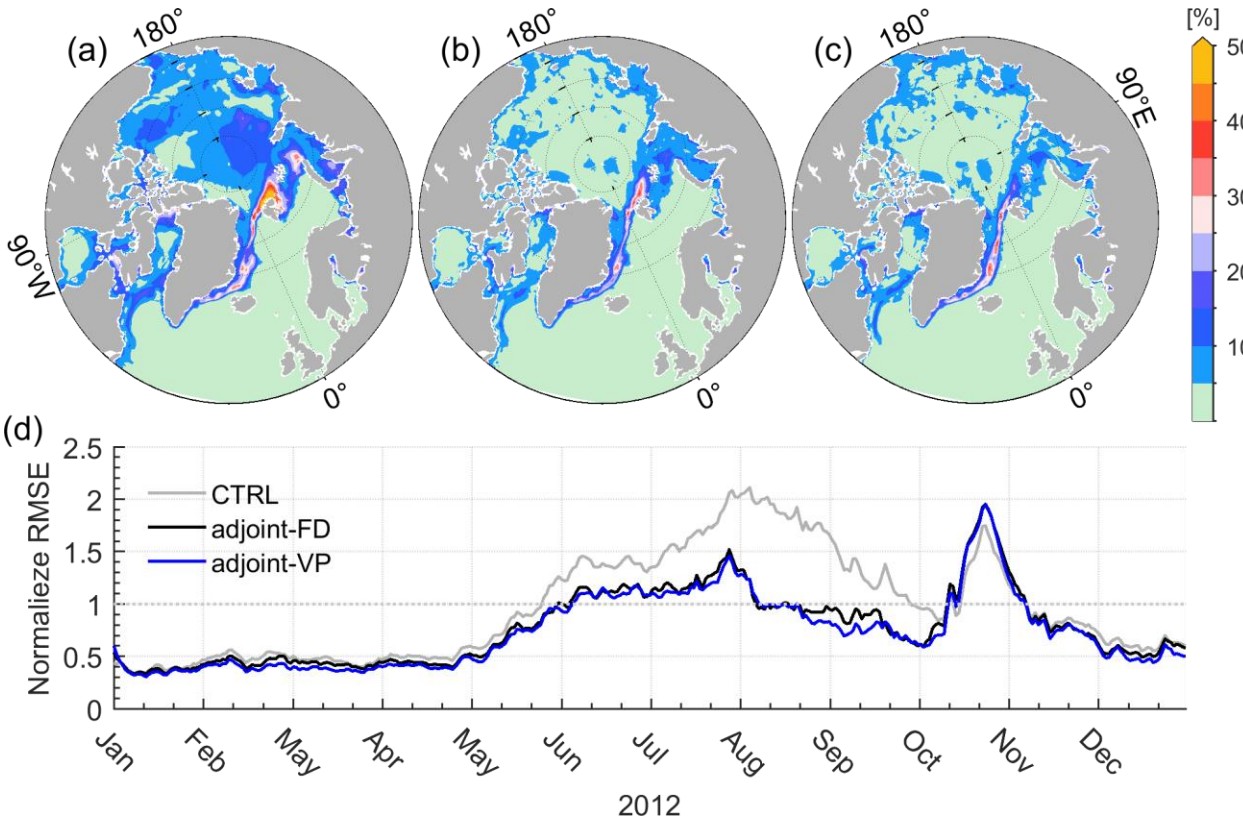


Figure 3. Root mean square errors (RMSEs) of SIC between the satellite measurements and (a) the control run, (b)
adjoint-FD, and (c) adjoint-VP averaged over 2012. Panel (d) shows the temporal variations in RMSEs normalised by
prior uncertainties in the three simulations averaged over the sea ice-covered regions.

The root mean square errors (RMSEs) of SIC averaged over 2012 (Figure 3a-c) and normalised by the prior
errors and averaged over the model domain (Figure 3d) show the geographical distribution and temporal evolution of
SIC errors, respectively. The normalised RMSEs in Figure 3d should be close to 1.0 if the optimisation found a model
simulation consistent with the observations and the prior uncertainties.
The control run (Figure 3a) shows pronounced RMSEs in the Beaufort Gyre (~15%), the central Eurasian Basin
(15%~20%), the marginal seas (15%~20%), and SIMs of the Atlantic sector (30%-50%). The normalised RMSEs
reveal that SIC errors remain small (~0.5) in the winter time (Figure 3d), indicating that the control run and the satellite
SIC measurements match well, but they grow quickly from May-September when the sea ice melts (Figure 3d).
Normalised RMSEs up to 1.5 are observed in October but quickly drop in November (Figure 3d). Therefore, SIC
errors are significant during the melting and refreezing periods (from May to November).
Both assimilation experiments reduce the SIC errors to less than 5% in the central Arctic Ocean and 10% in the
marginal seas. SIC errors of up to 20% persist in the Atlantic sector, where sea ice shows strong nonlinearity and the
tangent linear model can capture only part of the sea ice changes (APPENDIX B in Lyu et al., 2021a). Normalised
SIC errors from May to September are also reduced to close to 1.0 by assimilation of the daily SIC observations
(Figure 3d). However, SIC errors in October remain significant (Figure 3d) since the observed sea ice recovers much
faster than in the control run and the two assimilation runs (not shown here). This delayed sea ice recovery in the
model may be related to model uncertain parameters, such as the threshold thickness between thin and thick ice, which
determines the initial sea ice thickness formed in open water.

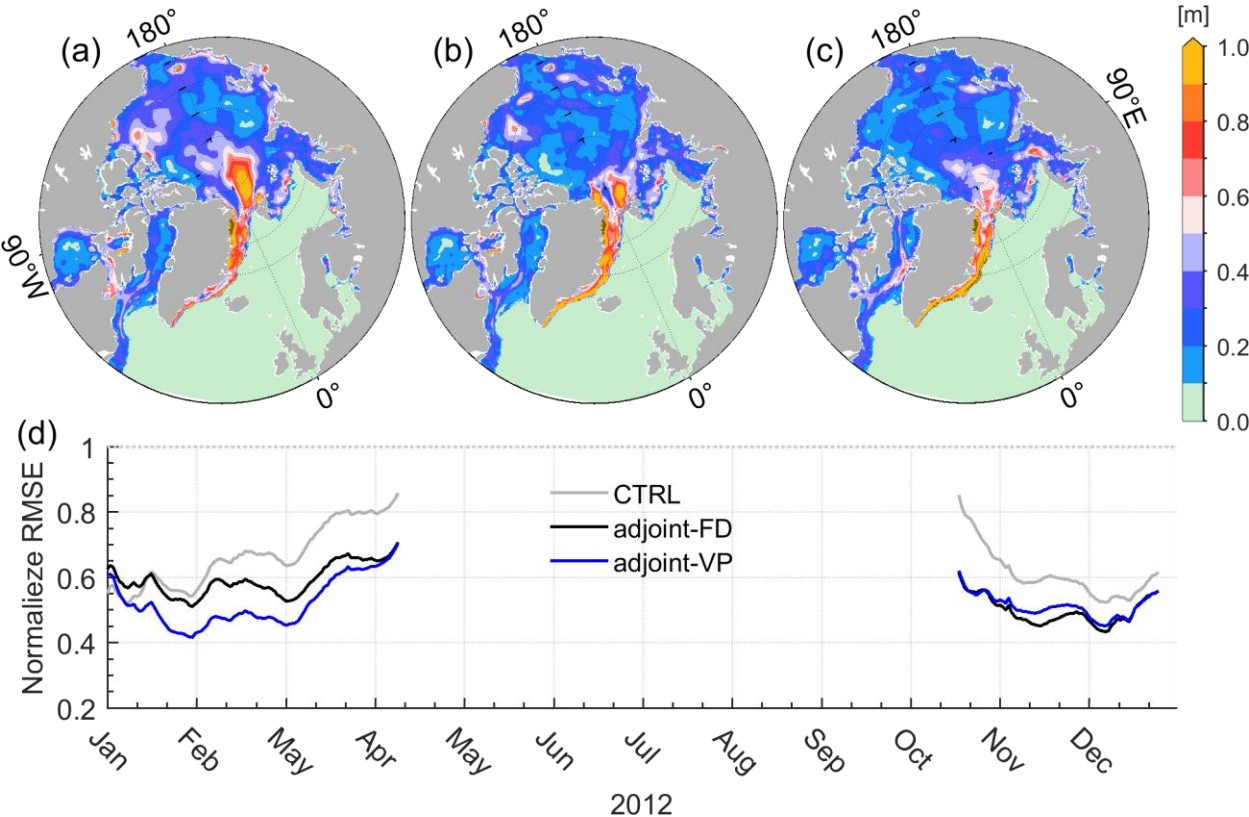

Figure 4. Root mean square errors (RMSEs) of SIT between the satellite measurements and (a) the control run, (b)
adjoint-FD, and (c) adjoint-VP averaged over 2012. Panel (d) shows the temporal variations in RMSEs normalised by
prior uncertainties in the three simulations averaged over the sea ice-covered regions.
The control run shows SIT errors of up to 1.0 m in regions north and south of the Fram Strait and approximately
0.4~0.7 m in the Beaufort Gyre. In the Beaufort Gyre, the SIT errors are reduced to less than 0.3 m in adjoint-VP
(Figure 4c) and approximately 0.3-0.5 m in adjoint-FD (Figure 4b). Similar to the SIC errors, SIT errors of up to 1.0
m remain along the East Greenland Current, which seems to increase in the two assimilation experiments. The
temporal evolutions of normalised RMSEs show that the SIT errors grow quickly from February to April (Figure 4d).
Both assimilation experiments reduce the SIT errors, especially in adjoint-VP from January to April (Figure 4d).
However, the normalised RMSEs of SIT averaged over the model domain remain smaller than 1.0 and seem to grow
during the melting season. Again, the normalized SIT errors of smaller than 1.0 indicates that SIT uncertainties are
too large, and more accurate SIT observations (e.g., half of the uncertainties) and SIT observations during the melting
season are required to facilitate a significant impact on the model simulation.
**3.2.2 BGEP Mooring Measurements**
Independent sea ice draft measured by up-looking-sonar (ULS) on the BGEP moorings ($M_a$, $M_b$, and $M_d$ in
Figure 1) is used to validate the simulated sea ice draft. The simulated snow depth ($d_{snow}$) and SIT ($d_{SIT}$) are used to
compute the sea ice draft following the methods of Tilling et al. (2018):
$$\text{draft} = \frac{\rho_i \times d_{SIT} + \rho_s \times d_{snow}}{\rho_w} \tag{9}$$

where $\rho_i$, $\rho_s$, and $\rho_w$ are the densities of the sea ice, snow, and water, respectively, and are set to 910.0, 330.0, and
1027.5 kg m$^{-3}$ , respectively, as in our model.

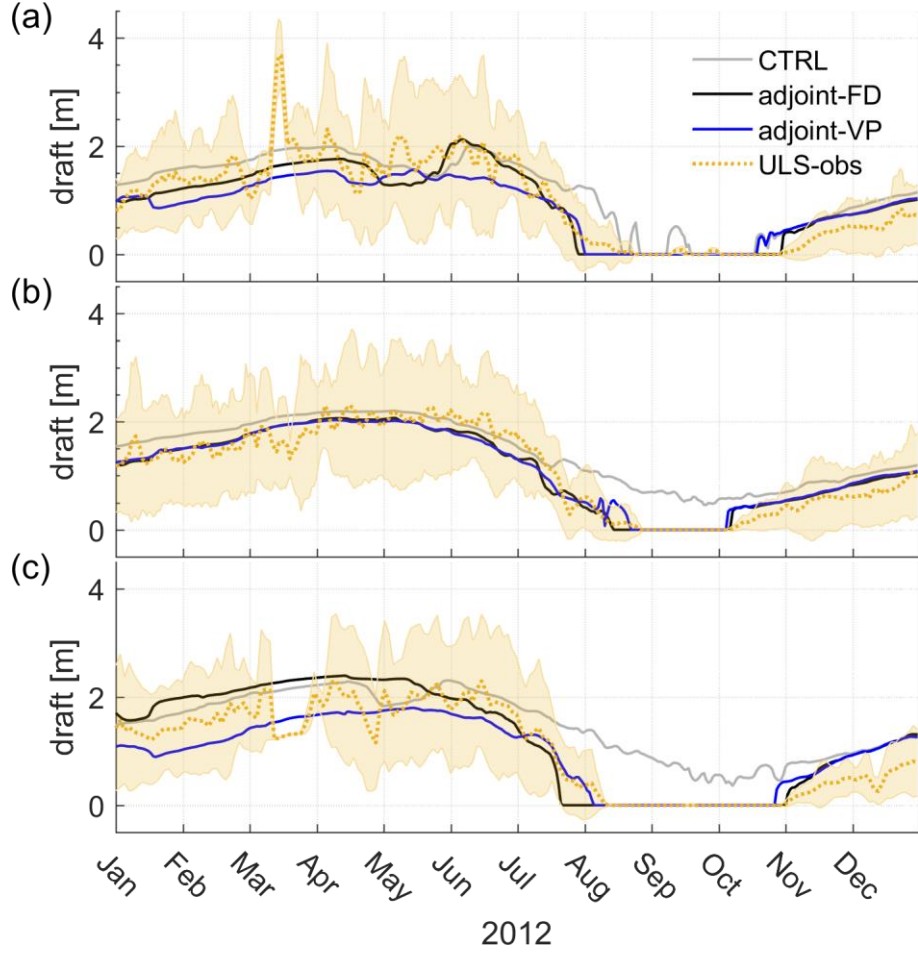


Figure 5. Daily time series of the sea ice draft (dotted yellow lines) and the daily standard deviation (shadings) at the
mooring locations (a) $M_a$, (b), $M_b$, and (c) $M_d$ compared with the control run and the two assimilation runs (see the
legend) throughout the year 2012. ULS-observed sea ice drafts are smoothed with a 5-day running average.

285   The ULS measurements depict stronger daily to sub-monthly sea ice draft variability than do the model
286  simulations, which may be related to ice floe motions. The control run simulates a delayed ice disappearance process
287  in $M_a$ (Figure 5a) and fails to reproduce the sea ice disappearance processes in $M_b$ (Figure 5b) and $M_d$ (Figure 5c) from
288  August to October. After optimisation, adjoint-VP and adjoint-FD reproduce the sea ice melting and refreezing
289  processes well, although errors of up to 0.5 m remain from January to June. Overall, the two assimilation runs
290  reproduce the local sea ice retreat and recovery process well.


292  **3.3 Ocean Temperature**

293   Ocean temperature changes are closely related to sea ice changes. Adjoint-VP introduces more pronounced ocean
294  temperature changes than does adjoint-FD. Here, we explore ocean temperature changes after assimilation.

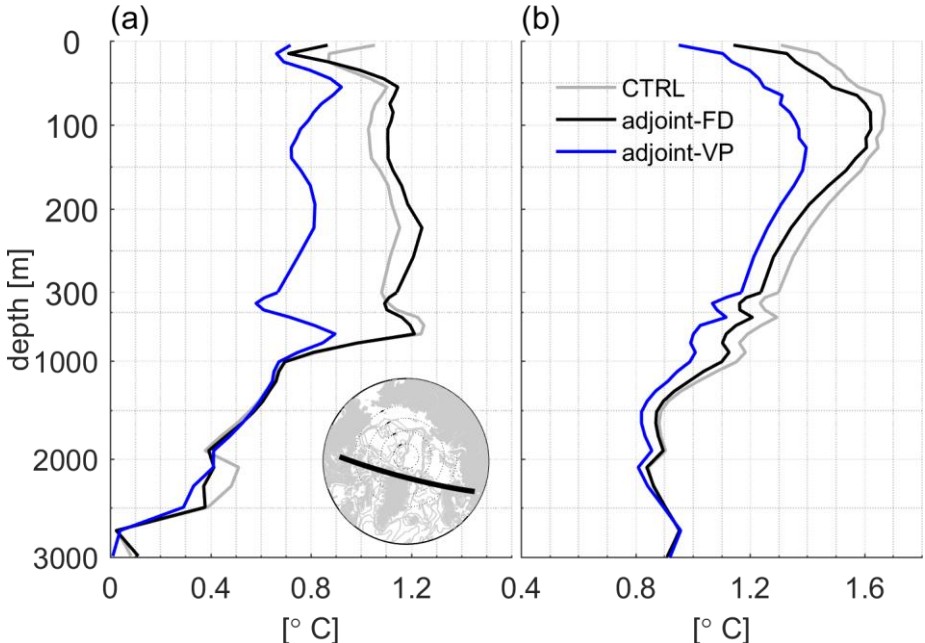

297 Figure 6. RMSEs of potential temperature (a) in the Arctic Ocean and (b) the North Atlantic Ocean in the three runs.
298 The Arctic Ocean and the North Atlantic Ocean are separated by the black lines in the bottom subplot.
300   In the Arctic Ocean, adjoint-FD reduces temperature errors only over the top 20 m, while adjoint-VP reduces
301  temperature errors up to 0.4 ℃ over the top 1000 m (Figure 6a). In the North Atlantic Ocean, adjoint-VP results a in
302  more pronounced RMSEs reduction up to 0.3 ℃ than adjoint-FD (Figure 6b).
303   Relative temperature error reductions over the top 50 m reveal an overall improvement in temperature with
304  occasional degradation (Figure 7a, b). Adjoint-VP results in a more significant error reduction than does adjoint-FD
305  in the North Atlantic Ocean (Figure 7a, b). In the southern Beaufort Gyre, the Laptev and Kara seas, and north of
306  Svalbard, both adjoint-VP and adjoint-FD increase the ocean temperature (over 50 m) since the two optimisation runs
307  reproduce the early retreat of the sea ice well, allowing more solar heating of the open water. In the North Atlantic
308  Ocean, adjoint-VP achieves more considerable temperature changes than does adjoint-FD both over the top 50 m
309  (Figure 7c, d) and from 50 m-700 m (Figure 7e, f). In the Arctic Ocean, adjoint-VP further introduces negative

temperature corrections between 50 and 700 m (Figure 7f), especially near the profile locations (see dots in Figure
7b).

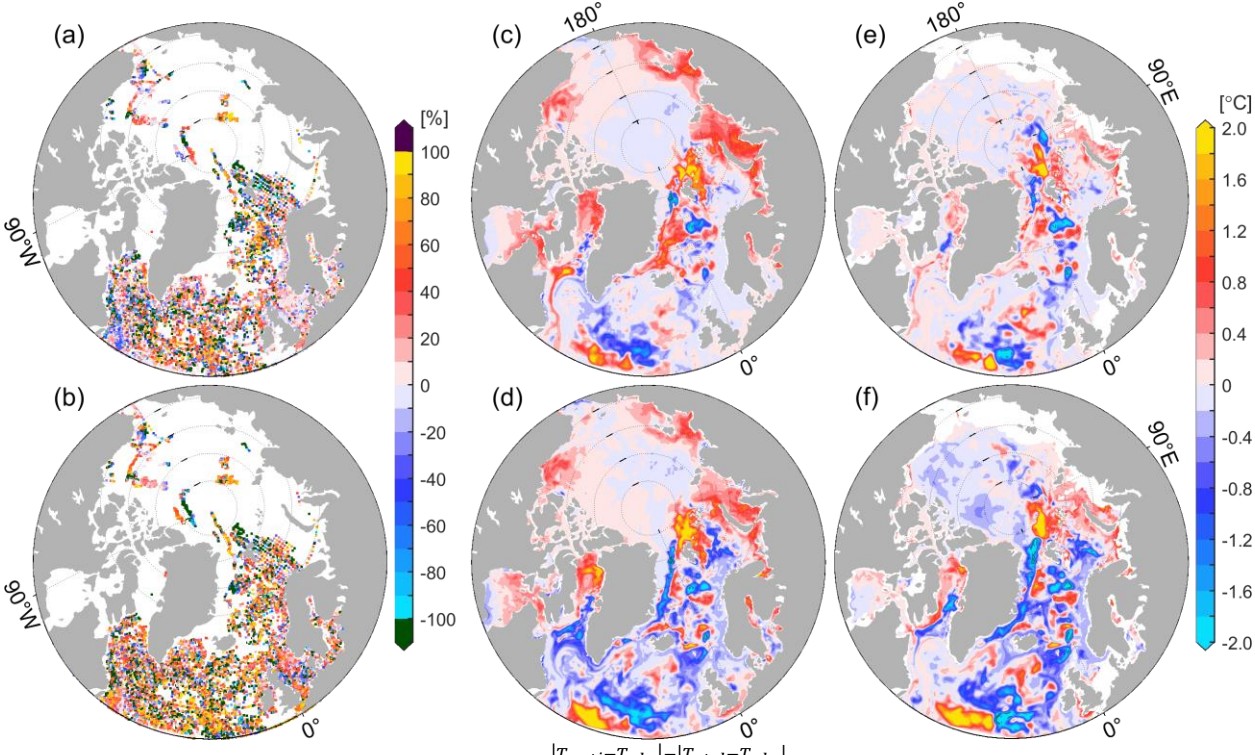

Figure 7. Relative temperature error reduction ($-\frac{|T_{opti}-T_{obs}|-|T_{ctrl}-T_{obs}|}{|T_{ctrl}-T_{obs}|} \times 100\%$) over the top 50 m at the profile
locations in (a) adjoint-FD and (b) adjoint-VP. Values >100% indicate over-adjustment. Panels (c) and (d) show the
temperature differences of adjoint-FD and adjoint-VP to the control run averaged over the top 50 m, respectively.
Panels (e) and (f) are the same as Panels (c) and (d), but for the 50-700 m layers.
In summary, adjoint-FD and adjoint-VP reproduce the SIC variations well in the Arctic Ocean, which further
reduces ocean temperature errors in the top layer by improving the atmosphere-ocean heat flux. Adjoint-VP achieves
more significant corrections to the ocean temperature over the open water and in the intermediate layer of the Arctic
Ocean than does adjoint-FD.
**4 Adjustment of the Control Variables**
The adjoint models project the model-data misfits onto the gradient of the objective function with respect to all
control variables simultaneously, which is used by the optimisation algorithm to adjust the control variables. In this
section, we compare adjustments of the control variables in the adjoint-FD and adjoint-VP and evaluate contributions
of individual adjustments of the control variables on the cost function reduction. We also compare the adjustments of
the control variables in adjoint-FD and adjoint-VP with differences between ERA5 (Hersbach et al., 2020) and NCEP-
RA1 reanalyses.
Among all the control variables, wind vectors and 2-m air temperature are considerably adjusted in adjoint-FD
and adjoint-VP but also show significant differences. In addition, adjoint-VP induces more pronounced adjustments
of the initital temperature and salinity than does adjoint-FD (not shown here). Here, we concentrate on the adjustments
of wind vectors and the 2-m air temperature.

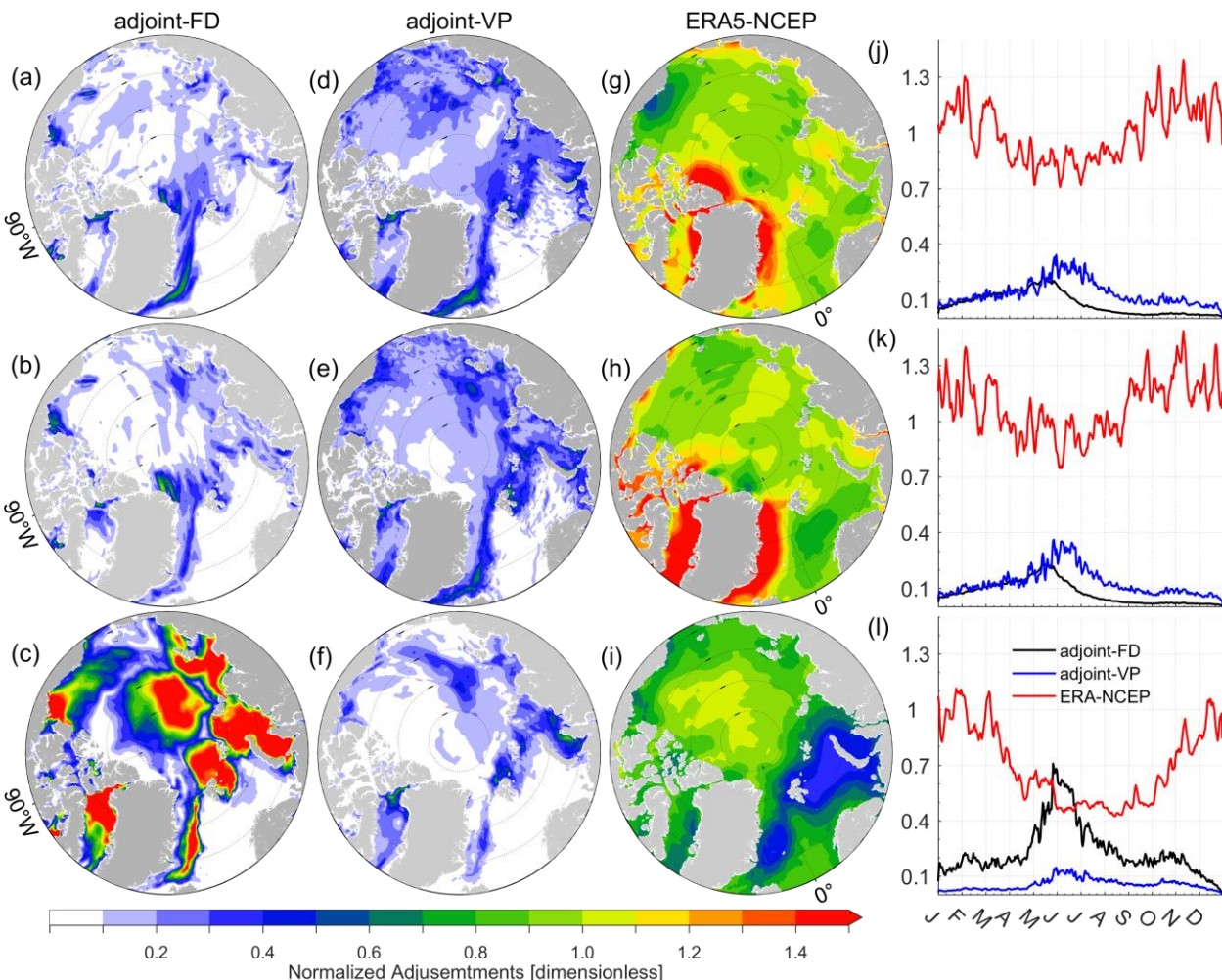

Figure 8. Root mean square (RMS) of the adjustments of the (a) wind *u*-component, (b) wind *v*-component, and (c) 2-m air temperature normalised by their prior uncertainties (dimensionless) in adjoint-FD and averaged over 2012. Panels (d)-(f) are similar to (a)-(c) but for adjoint-VP. Panels (g)-(i) show the normalised RMS of differences in the (g) wind *u*-component, (h) wind *v*-component, and (i) 2-m air temperature between ERA5 and NCEP-RA1 reanalyses (normalised by prior uncertainties in assimilation experiments). Panels (j)-(l) are the area averages of the normalised RMS of adjustments (differences) of the wind *u*-component, wind *v*-component, and 2-m air temperature (dimensionless) in adjoint-FD and adjoint-VP (ERA5-NCEP).

Figure 8 shows the root mean square (RMS) of the adjustments of the wind vectors and 2-m air temperature normalised by their prior uncertainties. The normalised RMS of the adjustments of the control variables should be smaller than 1.0 if the adjustments are within their prior uncertainties. Adjoint-FD slightly adjusts the wind vectors (with the normalised RMS of the adjustments being smaller than 0.4, Figure 8a, b), but the 2-m air temperature is significantly adjusted (with the normalised RMS of adjustments being greater than 1.5, Figure 8c). In adjoint-VP, the wind vectors and 2-m air temperature are slightly adjusted (Figure 8d-f) with their normalised RMS of adjustments being smaller than 0.3. In addition to the different amplitudes of the adjustments, the maximum adjustments of the wind vectors appear in June in adjoint-VP but in May in adjoint-FD (Figure 8j, k). Throughourt 2012, the 2-m air temperature is adjusted more prominently in adjoint-FD than in adjoint-VP (Figure 8l).

We note that the adjustments of the three atmosphere state variables in Figure 8a-f resemble the SIC (Figure 3a) and SIT (Figure 4a) error patterns, indicating that these adjustments are mostly determined by sea ice state errors that are projected on the control variables by the adjoint models rather than the background terms (the second and third terms on the right hand side of Equation (1)). Excluding the adjoint of sea ice rheology (adjoint-FD) results in over-adjustments of 2-m air temperature. With an approximated adjoint of sea ice rheology (adjoint-VP), we reduce the model-data misfits by slightly adjusting the control variables. The normalized adjustments of 0.1-0.6 indicate that the estimated prior uncertainties of atmospheric state remain too large.

Using the new generation ERA5 reanalysis, we further compare the ERA5-NCEP differences against the adjustments of the atmosphere state variables in terms of their spatial patterns and temporal variability. The ERA5 uses fractional SIC as surface boundary conditions, but NCEP-RA1 uses 0 and 1 for ice-free and ice-covered ocean, respectively. The purpose of this comparison is twofold: 1) it further justifies the rationale of the adjustment amplitudes, and 2) it examines whether the adjustments reflect the differences between the old generation NCEP-RA1 reanalysis and the new generation ERA5 reanalysis. For the wind vectors, the normalised RMS differences between the ERA5 and NCEP-RA1 reanalyses (Figure 8g, h) are much larger than the wind vector adjustments in adjoint-FD (Figure 8a, b) and adjoint-VP (Figure 8d, e). For the 2-m air temperature, the normalised ERA5-NCEP differences (>1.0, Figure 8i) are much larger than the normalised adjustments in adjoint-VP (<0.3, Figure 8f) but smaller than the normalised adjustments in adjoint-FD (>1.5, Figure 8c) in the Kara Sea, the Laptev Sea, the southern Beaufort Sea, the Eurasian Basin and the Makarov Basin. It is evident that the 2-m air temperature adjustments in adjoint-FD are too large. Averaged over the model domain and throughout 2012, the ERA5-NCEP differences are much larger than the adjustments (Figure 8j-l) in the two assimilation runs. In addition, the adjustments are larger from May to August than from September to April, while the ERA5-NCEP differences are larger in the winter season than in the summer season (Figure 8j-l). The comparisons confirm that the 2-m air-temperature is over-adjusted in adjoint-FD, especially from May to July (Figure 8l). The adjustments of wind vectors and 2-m air temperature do not resemble the ERA5-NCEP differences, indicating that the model-data differences cannot be fully fixed by replacing the old generation NCEP-RA1 reanalysis with the new generation ERA5 reanalysis.

**Table 3**. Contributions of the adjustments of 2-m air temperature, wind vectors, initial temperature and salinity (Initial T&S), and the remaining control variables (including initial mean SIT and SIC, 2-m specific humanity, precipitation, downwelling longwave, and net shortwave radiation) on the total cost reduction, SIC, SST, and temperature profiles in the two optimisation runs.

| | Adjoint-FD (%) | | | | Adjoint-VP (%) | | | |
|---|---|---|---|---|---|---|---|---|
| | 2-m air temperature | Wind vectors | Initial T & S | Remaining variables | 2-m air temperature | wind | Initial T & S | Remaining variables |
| $J_{total}$ | 29.0 | 17.5 | 6.0 | 3.0 | 5.3 | 52.6 | 25.1 | 5.0 |
| $J_{SIC}$ | 25.5 | 19.8 | 2.4 | 1.2 | 4.5 | 64.9 | 10.1 | 2.4 |
| $J_{SST}$ | 41.0 | 8.6 | 10.1 | 5.5 | 8.4 | 47.4 | 29.6 | 6.4 |
| $J_{prof\_T}$ | 3.9 | 4.9 | 4.3 | 4.3 | 4.4 | 40.9 | 182.0 | 7.9 |

By replacing the adjusted initial temperature and salinity, wind vectors, 2-m air temperature, and the remaining control variables with NCEP-RA1 datasets, we integrate the model and estimate the contributions of these variables

to the total cost reductions and individual components. Table 3 summarises contribution of individual control variables
to the total cost reductions and cost components of SIC, SST, and temperature profiles.
The small contributions of the adjustments of the remaining control variables ("Remaining variables" in Table 3)
to the cost function reductions in adjoint-FD and adjoint-VP highlight the importance of simultaneous adjustments of
the initial temperature and salinity, wind vectors and 2-m air temperature. In adjoint-FD, the adjustments of the 2-m
air temperature and wind vectors dominate the cost function reduction, especially the SIC components. In contrast,
adjoint-VP relies more on the adjustments of the wind vectors and the initial temperature and salinity. Besides, the
more pronounced ocean temperature improvements (see Figure 7) in adjoint-VP are mostly attributed to the
adjustments in the initial temperature and salinity (Table 3).
Overall, Adjoint-FD attributes more of the model-data misfits to the 2-m air temperature than does the adjoint-VP,
resulting in over-adjustments of the 2-m air temperature. By using an approximated adjoint of the sea ice rheology
(adjoint-VP), we reduce the model-data misfit by slightly adjusting the control variables. This leads to the conclusion
that the large 2-m air temperature adjustments in adjoint-FD is likely an overcompensation for wind errors that cannot
be appropriately corrected because of large errors in the respective cost function gradients.
**5 The Impacts on Sea Ice Retreat Processes**
A unique characteristic of the adjoint-based reanalysis is that its physical processes are described by the
governing equations of the model, allowing us to quantify the sea ice loss and the contributions of the sea ice dynamics
and sea ice-ocean-atmosphere fluxes through a closed budget analysis. In this section, we explore and compare the
mean SIT changes based on the model governing equation:
$$\frac{dh}{dt} = -\nabla \cdot (\vec{u}h) + F_{oi} + F_{ai} + F_{res} \tag{10}.$$

The mean SIT tendency $\frac{dh}{dt}$ is dominated by the sea ice advective flux ($-\nabla \cdot (\vec{u}h)$), ocean-sea ice heat flux ($F_{oi}$)
at the sea ice bottom, atmosphere-sea ice flux ($F_{ai}$) at the sea ice surface, and a residual term ($F_{res}$) including a snow
flooding effect and a source term to correct negative mean SIT to zero. $F_{oi}$ depends on ocean temperature difference
from freezing temperature (Maykut and Mcphee, 1995) and $F_{ai}$ consists of the radiation and turbulence fluxes over
the sea ice surface. The contributions of the residual terms are small and therefore we do not show them in the analysis
below.
Integrate the mean SIT over the Arctic Ocean (see Figure 9 for the locations), we derive Arctic sea ice volume
(SIV) changes. As shown in Figure 9a, the two assimilation runs change the total Arctic SIV changes in different ways.
Adjoint-VP reduces the Arctic SIV by reducing the initial Arctic SIV and changing the SIV tendency from May to
August. By September, adjoint-VP simulates more sea ice than the control run. Adjoint-FD slightly increases the
initial SIV, and the signals are invisible by February 2012. From May to July, adjoint-FD shows a stronger sea ice
melting process than the control run and adjoint-VP. By September, adjoint-FD simulates the most SIV among the
three simulations.
Based on Equation (10), we further compare SIV tendencies and the budget terms in the two assimilation runs
(Figure 9b). The two assimilation runs reveal that the seasonal SIV changes are dominated by $F_{ai}$ (magenta lines in
Figure 9b). Throughout the year, the ocean melts the sea ice from the bottom (blue lines in Figure 9b) and net sea ice
transport also reduces the Arctic sea ice (green lines in Figure 9b). However, we note that a much stronger sea ice loss
process occurs from May 20 to  June 15 in adjoint-FD (up to -193.0 km$^3$ day$^{-1}$) than in adjoint-VP (up to -125.0 km$^3$
day$^{-1}$), which is mainly attributed to $F_{ai}$ anomalies (magenta lines in Figure 9b).

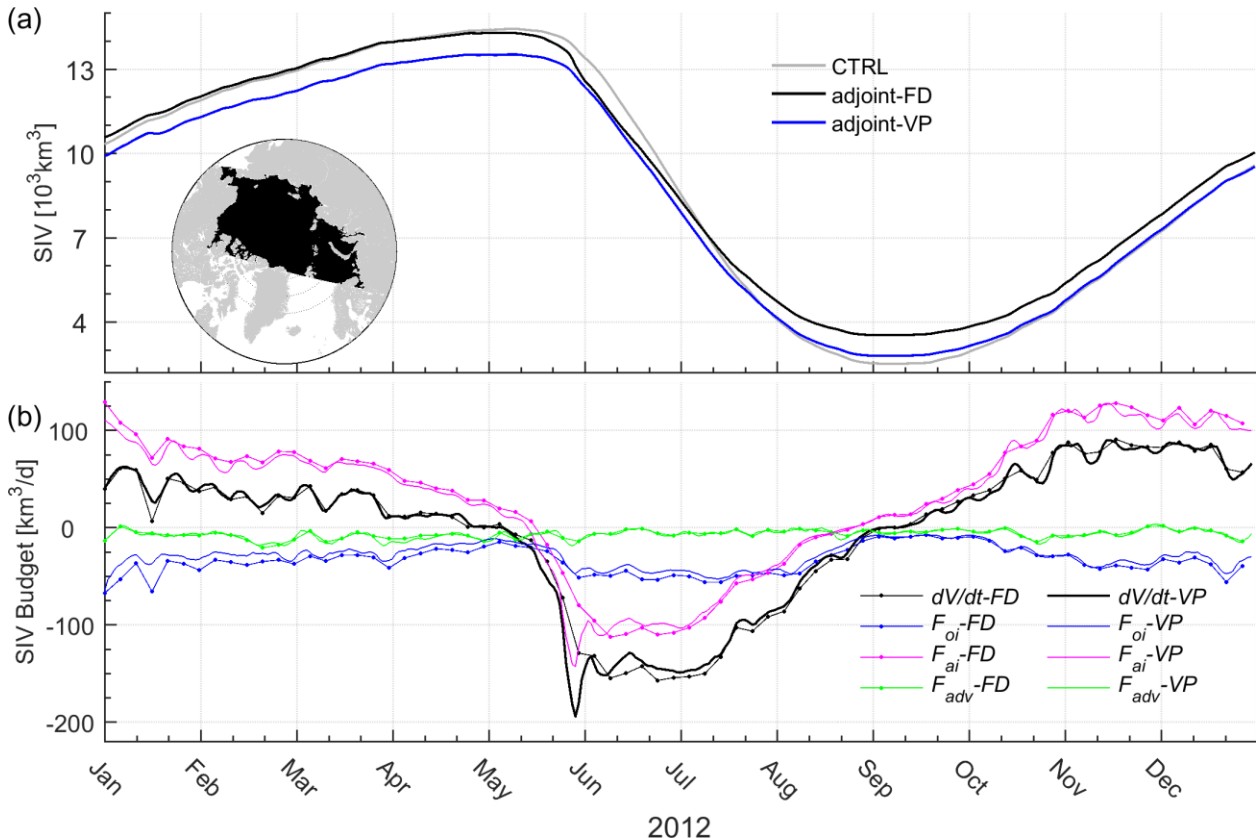

Figure 9. (a) SIV changes in the Arctic Ocean (see the bottom left subplot in Panel (a)) from January to December
2012. (b) SIV tendencies and contributions from $F_{oi}$, $F_{ai}$, and $F_{adv}$ in adjoint-FD and adjoint-VP (see the legend).

427        From May 20 to June 15, the Arctic Ocean observations rely most on satellite-measured SIC. Both the two

optimization runs reproduce the observed sea ice extents (SIEs, 15% SIC) well on June 15 (green and red lines in
Figure 10a, d), with adjoint-VP slightly better than adjoint-FD in the Barents and Kara Seas (Figure 10a, d).

430        On May 20, adjoint-FD simulates more sea ice than does adjoint-VP (Figure 9a). From May 20 to June 15,

adjoint-FD destroys the extra sea ice in the southeastern Beaufort Gyre, the Laptev Sea, the Kara Sea, and north of
Svalbard and Franz-Josef-Land through a stronger surface melting $F_{ai}$ (Figure 10b). At the same time, $F_{ai}$ results in
less sea ice loss (up to 0.6 m) in adjoint-FD in the central Arctic Ocean. Near the SIMs, $F_{adv}$ differences determine the
mean SIT differences (Figure 10a, c). In contrast, mean SIT differences from May 20 to June 15 between adjoint-VP
and the control run (Figure 10d) are mostly caused by $F_{adv}$ differences (Figure 10f) and $F_{ai}$ differences have little
contribution (Figure 10e), indicating that adjoint-VP modifies the SID to improve the model simulation during this
period.

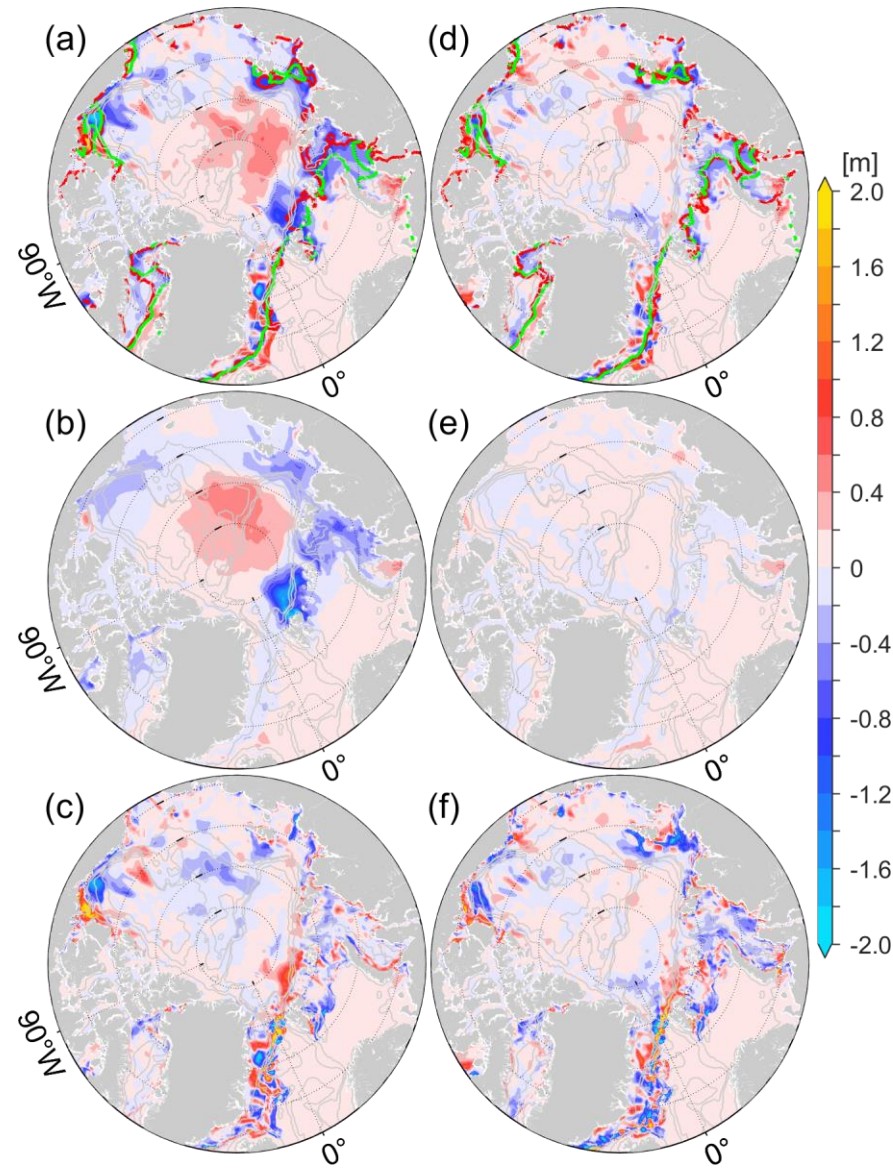

Figure 10. (a) Differences in $\int \frac{dh}{dt}$ integrated from May 20 to June 15 between adjoint-FD and the control run (adjoint-FD minus the control run), attributed to (b) $\int F_{ai}$ and (c) $\int F_{adv}$ differences. Panels (c)-(f) are the same as Panels (a)-(c) but for adjoint-VP. The red and green lines in Panels (a) and (d) indicate the model-simulated (a for adjoint-FD; d for adjoint-VP) and satellite-observed SIEs on June 15.

From May 20 to June 15, the significant sea surface melting anomalies (Figure 10b) are mainly caused by 2-m air temperature adjustments in adjoint-FD (Figure 11a). As shown, the 2-m air temperature is increased by more than 8 ℃ in the marginal seas (prior air temperature uncertainties are ~2-5 ℃) to facilitate the intense surface melting. In the central Arctic Ocean, the 2-m air temperature is reduced by 6 ℃ (Figure 11a), resulting in less sea ice loss up to 0.6 m (Figure 10b) than in the control run. In contrast, adjoint-VP adjusts the 2-m air temperature within ±3 ℃ in the marginal seas (Figure 11b), and the adjustments have little impact on the sea ice surface melting anomalies (Figure 10e). The 2-m air temperature differences averaged from May 20 to June 15 between the ERA5 and NCEP-RA1 reanalyses are within ±3 ℃ (Figure 11c), indicating that adjoint-FD over-adjusts the 2-m air temperature to destroy

the extra sea ice. Again, the spatial patterns 2-m air temperature adjustments in adjoint-FD and adjoint-VP don't
resemble that of ERA5-NECP differences.

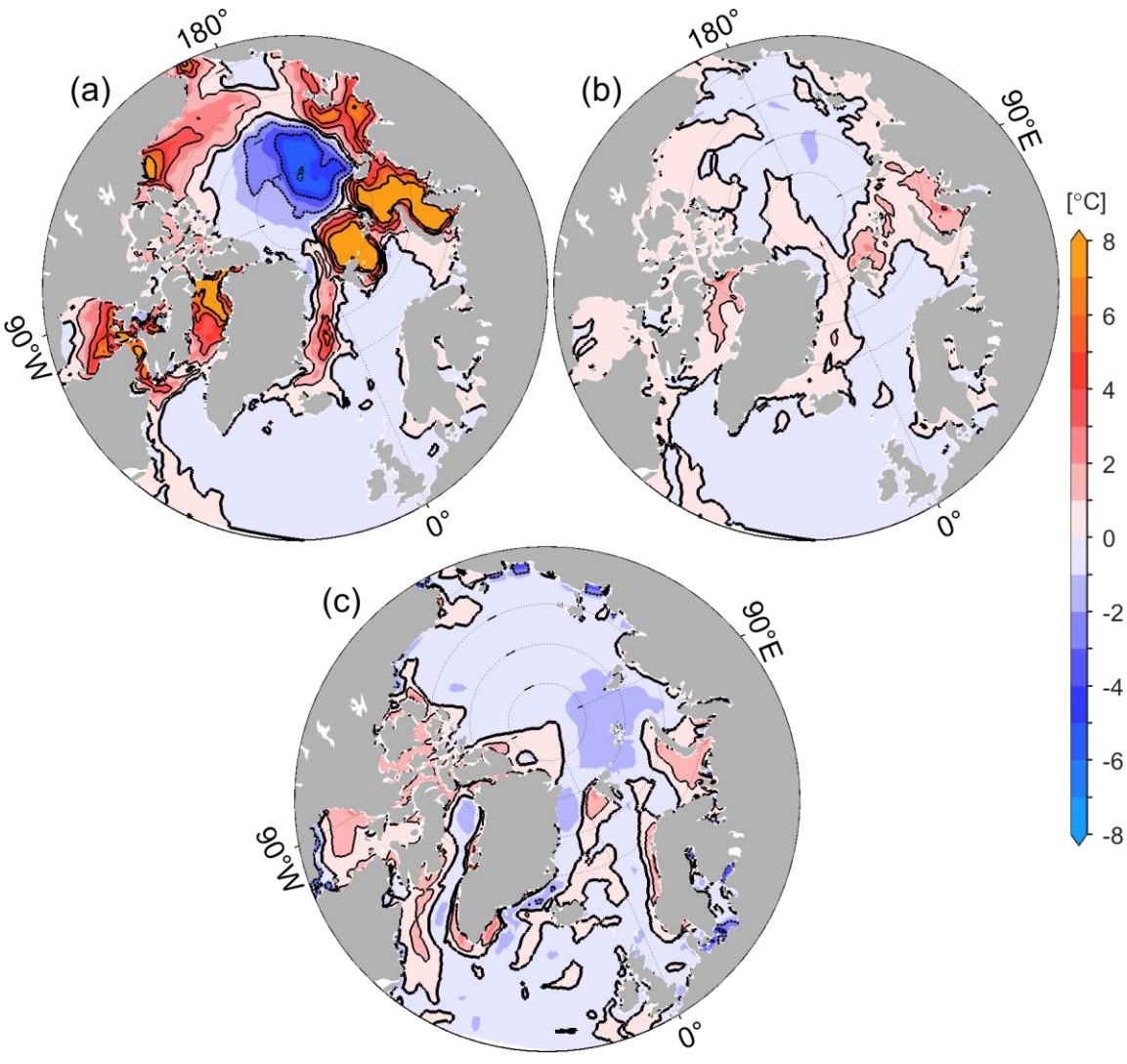


Figure 11. Adjustments of the 2-m air temperature averaged from May 20 to June 15, 2012, in (a) adjoint-FD and (b)
adjoint-VP. Panel (c) shows the 2-m air temperature differences between the ERA5 and NCEP-RA1 reanalyses
(ERA5 minus NCEP-RA1) averaged from May 20 to June 15, 2012. The contour intervals are 2 ℃.

In summary, the two optimisation runs successfully reproduce the sea ice retreat process in 2012 by assimilating
satellite and in situ measurements. However, the sea ice retreat processes differ in the two optimised simulations,
especially from May to June, when Arctic Ocean observations rely mostly on satellite-measured SIC. Considering the
amplitude of the 2-m air temperature adjustments, the adjustments of the control variables in adjoint-VP are more
reasonable than those in adjoint-FD due to the inclusion of the approximate adjoint of sea ice rheology.

## 6 Conclusions

The adjoint model is a powerful way to calculate the sensitivities of a target function to model variables and has been applied to coupled ocean and sea ice models for sensitivity studies (Heimbach et al., 2010; Kauker et al., 2009; Koldunov et al., 2013) and state estimations (Fenty and Heimbach, 2013; Koldunov et al., 2017; Lyu et al., 2021b; Nguyen et al., 2021). However, due to the persistent instability issues, the adjoint of sea ice dynamics are traditionally excluded or simplified to the adjoint of free-drift sea ice dynamics, which potentially hampers the accuracy of the coupled ocean and sea ice estimation.

Based on the study of Toyoda et al. (2019) and the coupled ocean and sea ice modelling and adjoint assimilation system (Lyu et al., 2021a), we approximate the adjoint of viscous-plastic sea ice dynamics and test the impacts on estimating the spatiotemporal variations in the Arctic ocean and sea ice state.

Two optimisations are performed, one including and one excluding the adjoint of sea ice rheology. Both assimilation exepriments reduce SIC and SIT errors and reproduce the sea ice retreat well. With the improved SIC retreat processes, adjoint-FD and adjoint-VP also show similar ocean temperature changes in the marginal seas and the southern Beaufort Gyre, as solar radiation heats the open water quickly as the sea ice retreats. With the improved adjoint of sea ice dynamics, adjoint-VP allows much stronger adjustments of the initial temperature, resulting in a more significant improvement on the temperature in the North Atlantic Ocean and the intermediate layer (50-700 m) of the Arctic Ocean.

Although that adjoint-FD computes much stronger sensitivities of the cost function to the wind vectors than does adjoint-VP, we note that adjoint-FD adjusts more (less) of the 2-m air temperature (wind vectors) than does adjoint-VP. It is evident that the adjoint sensitivities of wind vectors in adjoint-FD less efficiently reduce the cost function than those in adjoint-VP during the optimisation. Adjoint-FD strongly adjusts the 2-m air temperature to reduce the model-data misfits while adjoint-VP slightly adjusts all the control variables to improve the model simulation.

Using sea ice budget analysis, we further examine the sea ice retreat processes in adjoint-FD and adjoint-VP. We note that adjoint-FD and adjoint-VP show different sea ice thinning processes from May 20 to June 15 and in the marginal seas. Adjoint-FD destroys the extra sea ice in the marginal seas by substantially increasing the 2-m air temperature (up to 8 ℃), which is much larger than the ERA5-NCEP differences. In adjoint-VP, the sea ice thinning is moderate with more reasonable adjustments of 2-m air temperature (within ±3 ℃) and the size of the adjustments are much smaller than the ERA5-NCEP differences. Therefore, by including the adjoint of sea ice rheology, adjoint-VP projects the model-data misfits more properly to the control variables than that in adjoint-FD.

Parameter uncertainties significantly impact ocean and sea ice simulations (Lu et al., 2021; Massonnet et al., 2014; Sumata et al., 2019), and a lack of direct observations of key parameters potentially results in biases in the model simulations and predictions. The development of the adjoint of the viscous-plastic sea ice dynamics further introduces three parameters, including the ice compressive strength constant ($P^*$), ice strength decay constant ($C^*$),

and ratio of normal stress to shear stress (*e*), into the adjoint model. Since it remains unclear how well the tangent
linear approximation could represent the relations between the model parameters and the model state, in the future
studies, we will examine the accuracy of the adjoint sensitivities with respect to the model parameters and then further
improve the ocean-sea ice estimations by jointly estimating the state and parameters.
**7 Data availability**
The data used to create the plots in the paper are available at Pangaea (https://issues.pangaea.de/browse/PDI-33039).
Assimilated observations are listed in Table 1; and the up-looking sonar observed sea ice draft are from the Beaufort
Gyre Exploration Project (BGEP, https://www2.whoi.edu/site/beaufortgyre/).

*Author contribution.* G. Lyu designed the experiments, conducted the experiments and analysis. A. Koehl contributed
to the experiment design and interpretations. G. Lyu wrote the first draft. A. Koehl, D. Stammer, X. Wu, and M. Zhou
contributed to reviewing and editing the manuscript.

*Competing interests.* The authors declare that they have no conflict of interest.

**Acknowledgments**
This work was funded partly by the Open Fund Project of Key Laboratory of Marine Environmental Information
Technology, Ministry of Natural Resources of the People's Republic of China to G. Lyu and by the Shanghai Frontiers
Science Center of Polar Science (SCOPS) to M. Zhou. We thank NCEP and ECMWF for offering the NCEP/NCAR-
RA1 and ERA5 reanalyses. Thanks to ICDC at University of Hamburg and Alfred Wegener Institute for supplying
the ASI-SSM/I sea ice concentration and CryoSat-2/SMOS L4 datasets. We also acknowledge the Met Office, the
Copernicus Marine Service, and the Beaufort Gyre Exploration Project for archiving and sharing the EN4, the along-
track SLA , and sea ice draft datasets used in assimilation and independent validation.

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
