# Peer review of "2 3 Effects of including the adjoint sea ice rheology on estimating Arctic ocean-sea ice state 4 Guokun Lyu1, Armin Koehl2, Xinrong Wu3, Meng Zhou1,4, and Detlef Stammer2 5 1Shanghai Key Laboratory of Polar Life and Environment Scie"

_EGUsphere, 2022_

## Referee Comment (RC1)

***Effects of including the adjoint sea ice rheology on estimating
Arctic ocean-sea state***

by G. Lyu, A. Koehl, X. Wu, and D. Stammer

**Referee comment**

**GENERAL COMMENT:**

The manuscript explores the impact of improved approximation of the adjoint code for the visco-plastic (VP) sea ice model on the overall quality of the optimized solution. The authors obtain quantitative estimates of the improvement and demonstrate that better approximation of the adjoint velocity fields provides more realistic distribution of the optimal corrections among other components of the sea ice state. Although the manuscript is of certain interest to ice modeling community, it has a number of deficiencies that need to be addressed before publication. In particular, the authors need to better articulate the position of their Arctic ice modeling system among other Arctic ice modeling systems with data assimilation capabilities. Presentation of the material also needs substantial improvements: I would suggest editing the manuscript by a person with a better command of English.

**SPECIFIC COMMENTS:**

*Introduction:* the authors need to expand their overview of the data-driven ice modeling. In particular, they should discuss advantages of the adjoint/variational approach in forecasting seasonal changes of the sea ice state as compared to sequential methods (e.g., EnKF). Attention should also be paid to limited of differentiability of the VP rheology that imposes certain constraints on the utility of variational methods in ice forecasting.

*Section 2.1:*
   a) The utilized approximation to the adjoint of the tangent linear (TL) ice model should be described in more detail. Instability of the TL/adjoint codes of the Hibler's model is caused by violation of the positive semi-definite (PSD) property of the linearized C-grid discretization of the momentum equation, so it is instructive to give more details on what modifications of the code were made "to facilitate generation of the adjoint model". The cited paper by Losch et al (2010) is insufficient.
   b) A description of the "control run" (first mentioned in Table 2) is missing.
   c) Description of the assimilation window (line 100): It is unlikely that ice state in, say, December is controllable by the initial conditions on the 1st of January (Table 3). Provide more detail on the formulation of the assimilation window. Was it a sequence of twelve monthly windows (e.g., line 163)?

*Section 2.3:*
   a) Second term in eq, 3 is incorrect: check arguments of the viscosity coefficients (no commas between the indices), replace the deformation rate tensor by its trace

b) Lines 153-156: instability of the TL/adjoint code is caused by violation of the PSD property by the system solved by the LSOR method, and could be eliminated by using weak formulation of the ice momentum equation (Mehlmann & Richter, 2017). Besides, stabilization of the linearized rheology term could be achieved by augmenting of the exact adjoint code by Newtonian damping (Panteleev et al, 2021) rather than eliminating dependence of the viscosities on the deformation rates. In that respect, I would suggest to use the term "approximate adjoint" rather than "stabilized adjoint" throughout the text and add some discussion of the issue either here or in Section 6. Stabilization of an [exact] adjoint model implies adding extra terms for the purpose of moving the eigenvalues of the TL/adjoint matrices inside the unit circle.

c) Line 161: does "global mean" imply spatial *and annual* average in this context?

d) Line 166, 172 etc: "along the SIEs" SIE should be explained. If it stands for "sea ice extent", it is better to use "along the sea ice margins" (SIMs). Also, SIT and SIC should also be explained after their first appearance in the text.

*Section 3.1:*

a) Line 189: provide details on "could not be further reduced": does M1QN3 stall at a certain step magnitude during the line search? What gradient reduction was achieved in both the FD and VP cases at the convergence?

b) Lines 196-197: replacing "adjoint of the full sea ice dynamics" by "approximate adjoint of the sea ice rheology" would be more appropriate.

*Section 3.2:*

a) Line 222: why sea ice in the Atlantic sector exhibits stronger non-linearity? Explain.

b) Figure 4d: why there is a gap from May to October? No satellite data?

*Section 6:*

I would suggest to add a few computational quantities related to the improvement in the approximation of the adjoint model, such as the comparison of the cost function gradient decline curves with iterations, extra computational time (needed to execute 32 instead of 13 iterations), and extra CPU time/memory requirements related to the necessity of memorizing background viscosities and computing extra terms in the updated adjoint model. I would also add more discussion on the issues of instability and differentiability of the linearized VP rheology, and on the utility of the variational methods in optimizing ice conditions in general, especially in view of alternative rheologies (e.g. Mohr-Coulomb, elasto-brittle) based on more sophisticated models parameterizing sub-grid ice dynamics.

**TECHNICAL CORRECTIONS:**

There are numerous grammar and stylistic errors that should be corrected: To name a few:

L 47 and after: change hereafter to hereinafter.
L 61: "may overestimate", also change "towards" to "with respect to"
L174: change "contradicts the impacts" to "resist [or oppose] the impact"
L214: add "in Fig. 3d" after RMSEs
L222: change "exist" to "persist"
L226: unclear which "three simulations", change "may related" to "may be related"
L227-228: change to "threshold thickness… which triggers sea ice formation in open water"
L236 remove "larger"; change "constant" to "spatially uniform"
L293 change "which" to "but"

Figure 8: Color bar units? Both m/s and degrees Celsius? Comment in the caption.

L315: "By replacing [long list of variables] with their values and estimate [] contributions to cost reductions." Incomplete sentence. Split into several, clarify what you meant.

L319: variables

L323 "relies", "the wind vectors", "the temperature and salinity".

L326-330: Appears repetitive of the previous paragraph. Reformulate more clearly.

L362: move "thin**n**er" after "sea ice"

L363" "while sea ice in the Eurasian Basin and central Arctic is thicker by 0.4m"

L367: remove "thickened"

L420: change "seems" to "is evident"

L421: "is less efficient to reduce" to "are less efficient in reducing"

L429-430: change "which is melted" to "which destroys sea ice by"

L431: This statement looks trivial (it would be surprising if the effect were opposite).

---

## Referee Comment (RC2)

[referee-annotated manuscript omitted]

---

## Referee Comment (RC3)

Following is my review of the manuscript entitled "Effects of including the adjoint sea ice rheology on estimating Arctic ocean–ice state" by Guokun Lyu, Armin Koehl, Xinrong Wu, Meng Zhou, and Detlef Stammer (egusphere-2022-1099).

**General Comment**

In this study, motivated by Toyoda *et al.* (2019), the adjoint sea-ice model with viscous–plastic rheology (adjoint-VP) is applied to a coupled ocean and sea-ice state estimation system for the Arctic Ocean, and compared with the previous version in which the simplified adjoint sea-ice model of free drift (adjoint-FD) is used to avoid numerical instability. One year of optimization experiment for 2012 shows that the adjoint-VP can produce better state of the ocean and sea-ice through more appropriate dynamic and thermodynamic processes than the adjoint-FD.

Such findings are important for a further development of global-scale ocean state estimation and data assimilation studies, and could be worth to be published in *Ocean Science*. However, the manuscript has many deficiencies listed below and needs to be substantially revised before acceptance.

**Specific Comments**

1. Line 44: ECCO should be defined here.

2. Figure 1: It might be better to indicate important seas and straits.

3. Line 86: Describe the bulk formulae and related parameters used in this study, or cite appropriate references.

4. Explain why the open boundary conditions and the river runoff are not included in control variables.

5. Line 101: Explain what "effective" thickness means.

6. Line 118: BGEP should be defined here.

7. There are no explanations for SIC, SIE, SIT, SLA, and SST.

8. Section 2.3: Briefly describe the treatment of snow on sea ice, which may affect surface albedo and thermodynamic processes, or cite appropriate references.

9. Line 191: It is better to write explicitly that satellite-observes SST ($J_{SST}$) and SIC ($J_{SIC}$).

10. Line 196: It is misleading to call "the adjoint of full sea ice dynamics", because adjoint-VP still uses an approximated form of viscous–plastic rheology.

11. Section 3.1 and Table 2: The reviewer supposes that relative costs of individual constituents depend on their number of observations. If this is true, it might be better to indicate the total number of each measurement in Table 1.

12. Figure 3, caption: Explicitly mention that (a)–(c) are average of 2012.

13. Line 217: Explain what "sea ice extent regions" means.

14. Section 3.2.1: The normalized SIC errors of about 0.5 indicates that simulated SICs are overfitted to observations. Discuss this point.

15. Figure 4, caption: Describe the averaging period for (a)–(c).

16. Section 3.2.2: There are no description of Figure 5.

17. Figure 5, caption: Describe the averaging period.

18. Figures 3, 4, 5, and 6: It might be better to use the same colors among these figures for CTL, adjoint-FD, and adjoint-VP.

19. Line 337: Explain why April 10 and September 20 are chosen for this analysis.

20. Figure 9, caption: Explicitly mention that (a)–(e) are for the control run.

21. Figure 9: It seems that the red lines in (a), (f), and (k) are the September SIE from the control run, the black lines in (g)–(j) are from the adjoint-FD, and those in (k)–(o) are from adjoint-VP.

22. Line 369: It sounds strange that the SIC change through ice–albedo feedback is categorized as $F_{oi}$ rather than $F_{ai}$.

**Technical Corrections**

1. Use the same terminology throughout the manuscript. There are variants, e.g., "coupled ocean and sea-ice model", "coupled ocean and sea ice model", or "coupled ocean–sea ice model"; "Barents and Kara Seas" or "Kara and Barents Seas".

2. Lines 60 and 63: Dynamic should read dynamics.

3. Line 63: Zhang and Hibler Iii, 1997 should read Zhang and Hibler, 1997.

4. Line 108: $Q^{-2}$ should read $Q_a^{-2}$.

5. Line 125: 0.25% should read > 25%.

6. Equation 3: $\varepsilon_{i,j}$ should read $\varepsilon_{ij}$.

7. Line 152: $C^*$ should be 20.0 rather than –20.0.

8. Line 163: 31 January 31 should read January 31.

9. Line 180: Dynamic should read dynamics.

10. Line 195: $J_{sst}$ should read $J_{SST}$.

11. Line 207: Visual should read visible.

12. Line 274: Convert the second "In" to lowercase.

13. Line 374: Betterthan should read better than.

14. Line 374: Barent should read Barents.

15. Line 413: Incuded should read included.

16. Line 425: 20. September should read September 20.

17. Figures 1, 3, 4, and 7: Paint the Great Britain Island gray.

---

## Author Comment (AC1)

**Response to Reviewer**

By Guokun Lyu on behalf of all coauthors

We thank the reviewer for their careful reading of the manuscript and for their advice to improve the work. We have revised the manuscript according to the reviewer's recommendation. Below, we respond to the reviewer's questions and suggestions. Our responses are highlighted in blue.

**GENERAL COMMENT:**

The manuscript explores the impact of improved approximation of the adjoint code for the viscoplastic (VP) sea ice model on the overall quality of the optimized solution. The authors obtain quantitative estimates of the improvement and demonstrate that a better approximation of the adjoint velocity fields provides a more realistic distribution of the optimal corrections among other components of the sea ice state. Although the manuscript is of certain interest to ice modeling community, it has a number of deficiencies that need to be addressed before publication. In particular, the authors need to better articulate the position of their Arctic ice modeling system among other Arctic ice modeling systems with data assimilation capabilities. Presentation of the material also needs substantial improvements: I would suggest editing the manuscript by a person with a better command of English.

Response:

We thank the reviewer's suggestions. The complexity of this coupled ocean-sea ice model is similar to most of the current Arctic ocean-sea ice modeling and assimilation systems (see Table 1 in Uotila et al., 2019). The reconstructed sea ice and the mixed-layer sea water temperature and salinity in our previous Arctic reanalysis datasets are better than the TOPAZ4 and PIOMAS datasets (Lyu et al., 2021b). Section 2.1 explains more details about the thermodynamic-dynamic sea ice model. Besides, we edited the manuscript with native speakers.

**SPECIFIC COMMENTS:**

Introduction: the authors need to expand their overview of the data-driven ice modeling. In particular, they should discuss advantages of the adjoint/variational approach in forecasting seasonal changes of the sea ice state as compared to sequential methods (e.g., EnKF). Attention should also be paid to limited of differentiability of the VP rheology that imposes certain constraints on the utility of variational methods in ice forecasting.

Response:

We thank the reviewer's comments here. **We have introduced different data assimilation techniques in Paragraphs 3-4 in "1 Introduction".** In this part, we introduced the advantages/disadvantages of statistical-based and adjoint-based assimilation methods. Besides, we discussed the future developments and applications of the adjoint method in the coupled ocean and sea ice model and its potential difficulties in Section 6.

The TL/ADJ model is derived from the discrete coupled ocean and sea ice model. Besides "the limited differentiability of the VP rheology," other factors, including 1) nonlinearity vs

approximated tangent linear approximation; 2) modifications on the adjoint model to ensure the stability of the adjoint model (L180-184 in the manuscript); 3) the "if", "where" etc. statements in the discrete model, further hamper the usefulness/accuracy of the adjoint model. **We have examined how well the approximated TL/ADJ could predict error propagation in the nonlinear model (APPENDIX A in Lyu et al., 2021a, and our answer to Section 3.2 below).** Since we concentrate on approximating the adjoint of sea ice rheology in this study, we add paragraph 5 (L 59-66) to introduce the problems in adjoint of sea ice rheology in the current adjoint ocean and sea ice model, highlighting the necessity of this study.

Section 2.1:
a) The utilized approximation to the adjoint of the tangent linear (TL) ice model should be described in more detail. Instability of the TL/adjoint codes of the Hibler's model is caused by violation of the positive semi-definite (PSD) property of the linearized C-grid discretization of the momentum equation, so it is instructive to give more details on what modifications of the code were made "to facilitate generation of the adjoint model". The cited paper by Losch et al (2010) is insufficient.

Response:
We thank the reviewer's comment.

The MITgcm ocean model was developed with automatic differentiation software (Transformation of Algorithms in FORTRAN, TAF, Giering and Kaminski, 1998), which could automatically generate the adjoint codes based on the discrete nonlinear models. Losch et al. (2010) reformulated the dynamic-thermodynamic sea ice model from climate models on an Arakawa C grid to match the MITgcm oceanic grid. Further, **they modified many codes for efficient and accurate automatic differentiation using TAF.** The modifications are mainly on the forward model and tedious; therefore, we didn't explain more details. **We describe modifications of the adjoint codes in Section 2.3 (L180-184).**

b) A description of the "control run" (first mentioned in Table 2) is missing.
Response:
The control run is iteration 0. We have added a description of the control run in L215 (before Table 2).

c) Description of the assimilation window (line 100): It is unlikely that ice state in, say, December is controllable by the initial conditions on January 1 (Table 3). Provide more detail on the formulation of the assimilation window. Was it a sequence of twelve monthly windows (e.g., line 163)
Response:
We use **a 1-year assimilation window (the whole 2012 year)** in the two assimilation experiments. This large assimilation window is unique to the ECCO-like reanalysis. To achieve this large assimilation window (beyond the predictability of the nonlinear system), we have to modify the adjoint model by:
    1) disable the K-profiles mixing parameterization scheme
    2) Increase the Laplacian diffusivity of heat and salinity to 500 $m^2 \cdot s^{-1}$ and lateral eddy

viscosity to 10,000 $m^2 \cdot s^{-1}$;

    3) apply a spatial filter to sensitivity variables calculated in the adjoint of the thermodynamic sea ice model (see APPENDIX in (Lyu et al., 2021b) for detail)

    4) modified the adjoint of a VP sea ice dynamic (see section 2.3 of this manuscript).

We have added descriptions of these modifications explicitly in L180-184.

    It's true sea ice conditions on January 1 are unlikely impacts the sea ice state at the end of the year due to sea ice memory. However, the adjoint method also adjusted the atmospheric forcing throughout the assimilation window to reduce the model-data misfits.

**Section 2.3:**

a) Second term in eq, 3 is incorrect: check arguments of the viscosity coefficients (no commas between the indices), replace the deformation rate tensor by its trace

Response:

We thank the reviewer for pointing out the mistakes. We have revised these mistakes in the revised manuscript (L157).

b) Lines 153-156: instability of the TL/adjoint code is caused by violation of the PSD property by the system solved by the LSOR method, and could be eliminated by using weak formulation of the ice momentum equation (Mehlmann & Richter, 2017). Besides, stabilization of the linearized rheology term could be achieved by augmenting of the exact adjoint code by Newtonian damping (Panteleev et al, 2021) rather than eliminating dependence of the viscosities on the deformation rates. In that respect, I would suggest to use the term "approximate adjoint" rather than "stabilized adjoint" throughout the text and add some discussion of the issue either here or in Section 6. Stabilization of an [exact] adjoint model implies adding extra terms for the purpose of moving the eigenvalues of the TL/adjoint matrices inside the unit circle.

Response:

    We thank the reviewer's advice on using the terminology "an approximate adjoint of VP sea ice dynamic" and for suggesting the studies of Mehlmann & Richter (2017) and Panteleev et al (2021). In the revised manuscript, **we have modified the terminology "an exact adjoint of a VP sea ice rheology" to "an approximated adjoint of a VP sea ice rheology."**

    Besides the "violation of the PSD property by the system solved by the LSOR method", we explain more about the instabilities of the adjoint model generated based on the discrete ocean and sea ice models.

    In the atmosphere and ocean communities, previous studies have pointed out several reasons for the instabilities of the adjoint model:

    1) conflicts with Courant-Friedrichs-Lewy (CFL) conditions (Zhu and Kamachi, 2000) due to the discrete schemes; In this case, we have to reduce the model time step.

    2) linearizing the strong nonlinear processes such as KPP mixing parameterization, sea ice rheology, etc. Linearizing these processes will lead to a much faster exponential error growth (or eigenvalues) than the nonlinear model; We usually eliminate the dependence of the strong nonlinear parameterization on the model state (e.g., the viscosities on the deformation rates in our model).

    3) integrate the adjoint beyond the predictability limits of the nonlinear system; This

problem can be solved by adding additional terms to move the eigenvalues of adjoint matrices inside the unit circle, as the reviewer suggested.

As mentioned in the manuscript, the ocean and sea ice model's adjoint has suffered from instability for a long time when we include the adjoint of sea ice rheology. Toyoda et al. (2019) suggested that "the dependence of the viscosities on the deformation rates" results in strong nonlinearities (the second cause mentioned above), and eliminating their dependence could remove the abnormal positive engine value of the adjoint model. Therefore, we examine whether it works in a VP rheology and test the impacts on estimating the Arctic ocean and sea ice state.

c) Line 161: does "global mean" imply spatial *and annual* average in this context?
Response:
**"global mean" indicates the average over the model domain at the current model time step**. We can only use the mean adjoint sensitivities of sea ice velocity when applying this filtering process at the present step.

d) Line 166, 172 etc: "along the SIEs" SIE should be explained. If it stands for "sea ice extent", it is better to use "along the sea ice margins" (SIMs). Also, SIT and SIC should also be explained after their first appearance in the text.
Response:
We thank the reviewer's comments. We have revised the related abbreviations in the context.

**Section 3.1:**
a) Line 189: provide details on "could not be further reduced": does M1QN3 stall at a certain step magnitude during the line search? What gradient reduction was achieved in both the FD and VP cases at the convergence?
Response:
We have provided more details on the optimization "In adjoint-FD and adjoint-VP, the optimizations stall at iterations 13 and 32, and the further cost function reductions at the last two successive iterations are 0.7% and 0.2% of the total cost, respectively. After the optimizations, the total cost and norms of the gradients are reduced by 32.3% and 59.2% in adjoint-FD and 40.2% and (89.3%) in adjoint-VP." (L215-218)

b) Lines 196-197: replacing "adjoint of the full sea ice dynamics" by "approximate adjoint of the sea ice rheology" would be more appropriate.

Response:
We thank the reviewer's advice. We have revised "adjoint of the full sea ice dynamics" to "approximate adjoint of the sea ice rheology" throughout the manuscript.

**Section 3.2:**
a) Line 222: why sea ice in the Atlantic sector exhibits stronger nonlinearity? Explain.
Response:
We thank the reviewer's question. This statement, "where sea-ice shows strong

nonlinearity and the tangent linear model could only capture part of the sea-ice changes (Appendix B in Lyu et al., 2021a)." comes from our previous research.

To evaluate how well the tangent linear model could predict the nonlinear errors, we performed three model simulations: 1) $F(C_{atm}(t))$ with default atmosphere forcing; 2) add perturbations $+\Delta C_{atm}(t)$ to atmosphere forcing $F(C_{atm}(t) + \Delta C_{atm}(t))$; 3) add perturbations $-\Delta C_{atm}(t)$ to atmosphere forcing $F(C_{atm}(t) - \Delta C_{atm}(t))$. The three model simulations are integrated for one year, and we evaluated the linear and higher-order components of the responses following the Taylor expansions:

$$\Delta F_1 = F(C_{atm}(t) + \Delta C_{atm}(t)) - F(C_{atm}(t)) = \frac{\partial F}{\partial C} \cdot \Delta C_{atm}(t) + o(\Delta C_{atm}(t)) \quad (A1)$$

$$\Delta F_2 = F(C_{atm}(t) - \Delta C_{atm}(t)) - F(C_{atm}(t)) = -\frac{\partial F}{\partial C} \cdot \Delta C_{atm}(t) + o(-\Delta C_{atm}(t)) \quad (A2)$$

where the linear and nonlinear error components are obtained by $\frac{1}{2}(\Delta F_1 - \Delta F_2)$ and $\frac{1}{2}(\Delta F_1 + \Delta F_2)$, respectively. By integrating the tangent linear model with the same atmospheric forcing perturbations $\Delta C_{atm}(t)$, we get the error evolution for the model variables.

Figure R1 reveals that much stronger high-order signals exist in the Atlantic sector in both the melting season (Figure R1b) and the freezing season (Figure R1e). The tangent linear model could represent the linear signal well in the melting season (Figure R1a, c) but its accuracy is much lower in the freezing season (Figure R1d,f). However, the causes for these differences need to be further investigated.

[Figure]

Figure R1. Linear (a) and nonlinear (b) components of the SIC changes averaged over May-June computed based on equations (A1)-(A2). Panel (c) is SIC changes predicted by the

approximated tangent linear model in May-June. Panels (d)-(e) are the same as panels (a)-(c), except that they are averaged over November-December. Adopted from Lyu et al. (2021a).

b) Figure 4d: why there is a gap from May to October? No satellite data?

Response:
From April 15 to October 15, the satellite cannot observe sea ice thickness.

**Section 6:**
I would suggest to add a few computational quantities related to the improvement in the approximation of the adjoint model, such as the comparison of the cost function gradient decline curves with iterations, extra computational time (needed to execute 32 instead of 13 iterations), and extra CPU time/memory requirements related to the necessity of memorizing background viscosities and computing extra terms in the updated adjoint model. I would also add more discussion on the issues of instability and differentiability of the linearized VP rheology, and on the utility of the variational methods in optimizing ice conditions in general, especially in view of alternative rheologies (e.g. Mohr-Coulomb, elasto-brittle) based on more sophisticated models parameterizing sub-grid ice dynamics.

Response:

   We thank the reviewer's comments here. **We have added the cost and gradients reductions in adjoint-FD and adjoint-VP in L215-218**. **Including the adjoint of sea ice rheology, we note that the adjoint model requires ~1.2 times of using the adjoint of a free-drift sea ice model (L187-188).** Further, we need to memorize ice velocity during the iterative solving processes, and the other variables are recomputed. For our configurations (480×416 horizontal grids ×50 levels) with a one-year assimilation window, one iteration (forward+adjoint run) requires 24 hour*208 CPU times.

   In this study and our previous study (Lyu et al.,2021a, b), we have tested the performance of the adjoint method in optimizing the initial and atmosphere conditions. In Section 6, we further discuss the development of optimizing model states and uncertain parameters based on the adjoint model. Once the forward sea ice model is developed with other more sophisticated sea ice rheologies, we believe the major problem is how to approximate (stabilize) the adjoint of sea ice rheology. However, since we are not sure about the nonlinearity and computational requirements of more sophisticated sea ice rheologies, we refrain from further discussion on the choice of alternative sea ice rheologies.

**TECHNICAL CORRECTIONS:**
There are numerous grammar and stylistic errors that should be corrected: To name a few:
Response: We thank the reviewer's comments and have corrected the manuscript's grammar and stylistic errors.
L 47 and after: change hereafter to hereinafter.
Response: we have changed "hereafter" to "hereinafter" throughout the manuscript (e.g., L187-

188).

L 61: "may overestimate", also change "towards" to "with respect to"
Response: We have rephrased these words.

L174: change "contradicts the impacts" to "resist [or oppose] the impact"
Response: we have changed the word "contradicts" to "oppose" (L207).

L214: add "in Fig. 3d" after RMSEs
Response: We added "in Fig. 3d" after RMSEs (L243).

L222: change "exist" to "persist"
Response: we have changed "exist" to "persist" (L252).

L226: unclear which "three simulations", change "may related" to "may be related"
Response:
The three simulations represent "the control run" and "the two assimilation runs". We have changed the words "three simulations" to "the control run, and the two assimilation runs". We also changed "may related" to "may be related" (L256-257).

L227-228: change to "threshold thickness… which triggers sea ice formation in open water"
Response: We thank the reviewer's comments. We have changed "demarcation" to "threshold" (L257).

L236 remove "larger"; change "constant" to "spatially uniform"

Response: we have removed "larger"; In this study, we used spatial-varying ice thickness errors. For accuracy, we have deleted these words.

L293 change "which" to "but
Response: We thank the reviewer's comment. We have changed "which" to "but" (L330).

Figure 8: Color bar units? Both m/s and degrees Celsius? Comment in the caption.
Response: Units in the color bars are dimensionless since the adjustments are normalized by their prior uncertainties. We have labeled the color bar with "dimensionless" and commented in the Figure 8 caption.

L315: "By replacing [long list of variables] with their values and estimate [] contributions to cost reductions." Incomplete sentence. Split into several, clarify what you meant.
Response:
We thank the reviewer's suggestion. We have rewritten these words to "By replacing the adjusted initial temperature and salinity, wind vectors, 2-m air temperature, and the remaining control variables with NCEP/RA1 datasets, we integrate the model and estimate their contributions to the total cost reductions and individual components" (L380-383). We test the relative contributions of individual control variables to cost function reduction.

L319: variables

Response:

We thank the reviewer for pointing out the mistakes and have deleted the duplicated "variables" (L385).

L323 "relies", "the wind vectors", "the temperature and salinity".

Response: We have corrected these mistakes here (L388).

L326-330: Appears repetitive of the previous paragraph. Reformulate more clearly.

Response:

We have rewritten the paragraph to "Overall, adjoint-FD attributes model-data misfits more on 2-m air temperature than adjoint-VP, resulting in over-adjustments of 2-m air temperature. By using an approximated adjoint of the sea ice rheology (adjoint-VP), we reduce the model-data misfits by slightly adjusting the control variables. This leads to the conclusion that the large 2-m air temperature adjustments in adjoint-FD are likely an overcompensation for wind errors that could not be corrected appropriately because of large errors in the respective cost function gradients." (L391-395).

L362: move "thinner" after "sea ice"

L363" "while sea ice in the Eurasian Basin and central Arctic is thicker by 0.4m"

L367: remove "thickened"

Responses: we thank the reviewer's corrections and have rewritten the paragraphs and corrected the related errors (L408-420).

L420: change "seems" to "is evident"

Response: we have changed the words "seems" to "is evident" (L484).

L421: "is less efficient to reduce" to "are less efficient in reducing"

Response: We thank the reviewer's comment and have changed the words to "less efficiently reduce " (L484).

L429-430: change "which is melted" to "which destroys sea ice by"

Response: We thank the reviewer's correction, and we have changed "which is melted" to "which destroys sea ice by" (L489).

L431: This statement looks trivial (it would be surprising if the effect were opposite)

Response:

We thank the reviewer's comments. **Including the adjoint of sea ice rheology is deemed to improve the accuracy of the adjoint ocean-sea ice model if we have no further modification on the adjoint model.** However, we further enhanced viscosity and diffusivity (or added additional viscosity and diffusivity) in the adjoint model to stabilize the adjoint model beyond the predictability limit. It is unclear whether the enhanced viscosity and diffusion in the adjoint model could damp out the signals introduced by the adjoint of sea ice rheology. This

study confirms that including this approximated sea ice rheology improves the ocean and sea ice estimation by adjusting the control variable slightly.

Reference:

[1] Lyu, G., Koehl, A., Serra, N., and Stammer, D.: Assessing the current and future Arctic Ocean observing system with observing system simulating experiments, Quarterly Journal of the Royal Meteorological Society, 147, 2670-2690, https://doi.org/10.1002/qj.4044, 2021a.

Lyu, G., Koehl, A., Serra, N., Stammer, D., and Xie, J.: Arctic ocean–sea ice reanalysis for the period 2007–2016 using the adjoint method, Quarterly Journal of the Royal Meteorological Society, 147, 1908-1929, https://doi.org/10.1002/qj.4002, 2021b.

Uotila, P., Goosse, H., Haines, K., Chevallier, M., Barthélemy, A., Bricaud, C., Carton, J., Fučkar, N., Garric, G., Iovino, D., Kauker, F., Korhonen, M., Lien, V. S., Marnela, M., Massonnet, F., Mignac, D., Peterson, K. A., Sadikni, R., Shi, L., Tietsche, S., Toyoda, T., Xie, J., and Zhang, Z.: An assessment of ten ocean reanalyses in the polar regions, Climate Dynamics, 52, 1613-1650, 10.1007/s00382-018-4242-z, 2019.

Zhu, J. and Kamachi, M.: The role of time step size in numerical stability of tangent linear models, Monthly Weather Review, 128, 1562-1572, 2000.

---

## Author Comment (AC2)

**Response to Referee**

Following is my review of the manuscript entitled "Effects of including the adjoint sea ice rheology on estimating Arctic ocean–ice state" by Guokun Lyu, Armin Koehl, Xinrong Wu, Meng Zhou, and Detlef Stammer (egusphere-2022-1099).

General Comment

In this study, motivated by Toyoda *et al.* (2019), the adjoint sea-ice model with viscous–plastic rheology (adjoint-VP) is applied to a coupled ocean and sea-ice state estimation system for the Arctic Ocean, and compared with the previous version in which the simplified adjoint sea-ice model of free drift (adjoint-FD) is used to avoid numerical instability. One year of optimization experiment for 2012 shows that the adjoint-VP can produce better state of the ocean and sea ice through more appropriate dynamic and thermodynamic processes than the adjoint-FD.

Such findings are important for a further development of global-scale ocean state estimation and data assimilation studies, and could be worth to be published in *Ocean Science*. However, the manuscript has many deficiencies listed below and needs to be substantially revised before acceptance.

Response:

We thank the reviewer's time for reviewing the manuscript and helping us to improve the manuscript. We have revised the manuscript following the reviewer's suggestions. Our responses to the reviewer's comments are listed below.

Specific Comments

1. Line 44: ECCO should be defined here.

Response:

We thank the reviewer's comment and have defined ECCO here (L51).

2. Figure 1: It might be better to indicate important seas and straits.

Response: We thank the reviewer's suggestion. We have labeled the major basins and straits in Figure 1 and described them in the caption.

3. Line 86: Describe the bulk formulae and related parameters used in this study, or cite appropriate references.

Response:

We thank the reviewer's advice. Surface fluxes computations in MITgcm are based on the bulk parameterization of Large and Yeager (2004). We use their default parameter values. In the manuscript, we added a reference here (L96).

4. Explain why the open boundary conditions and the river runoff are not included in the control variables.

Response:

We thank the reviewer's comment. The choice of the control variables is for reasons of "simplicity and the robust performance of this coupled data assimilation system." Since

we concentrate on developing the adjoint of sea ice rheology, we think using a configuration that works well now is safe. Therefore, we use our previous setup. In the future development of Arctic reanalysis, we will include river runoff, the open boundary conditions, and parameters in the control variables. In the manuscript, we explain the reasons in L122-128.

5. Line 101: Explain what "effective" thickness means.
Response:
We thank the reviewer's comment. In the model formulation, **"effective ice thickness"**

**means "sea ice thickness multiply sea ice concentration" or "volume per unit area."**

In L87, we have defined it to "mean sea ice thickness (in volume per unit area, mean SIT hereinafter)" and used the terminology "mean SIT" throughout this manuscript.

6. Line 118: BGEP should be defined here.
Response:
We have defined BGEP in L135.

7. There are no explanations for SIC, SIE, SIT, SLA, and SST.

Response:

We thank the reviewer for pointing out these mistakes. We have added the full names of SIC (L86), SIE (L428), SIT (L87), SLA (L143), and SST (L145) at their first appearance.

8. Section 2.3: Briefly describe the treatment of snow on sea ice, which may affect surface albedo and thermodynamic processes, or cite appropriate references.
Response:
We thank the reviewer's comment. A diagnostic snow model is applied on sea ice, which modifies the heat flux and surface albedo. In the revised manuscript, we describe the snow model in Section 2.1 and cite the related reference (L84-87).

9. Line 191: It is better to write explicitly that satellite-observes SST ($J_{SST}$) and SIC ($J_{SIC}$).
Response:
We thank the reviewer's comment. We have written the "satellite-observed SST ($J_{SST}$) and SIC ($J_{SIC}$)" explicitly (L221).

10. Line 196: It is misleading to call "the adjoint of full sea ice dynamics", because adjoint-VP still uses an approximated form of viscous–plastic rheology.
Response:

We thank the reviewer's comment on "approximated form of viscous–plastic rheology." ". In this manuscript, we change "adjoint of VP rheology" to "approximated adjoint of VP rheology" (L212).

11. Section 3.1 and Table 2: The reviewer supposes that relative costs of individual constituents depend on their number of observations. If this is true, it might be better to indicate the total number of each measurement in Table 1.
Response:
We thank the reviewer's comment. We have added the number of individual observation types in Table 1.

12. Figure 3, caption: Explicitly mention that (a)–(c) are average of 2012
Response:
We thank the reviewer's suggestion. We have included "averaged over 2012" in the Figure 3 caption.

13. Line 217: Explain what "sea ice extent regions" means.

Response:

We thank the reviewer's comment. Here, it means "sea ice margins (SIMs)". We have revised it to "sea ice margins (SIMs)" in L199.

14. Section 3.2.1: The normalized SIC errors of about 0.5 indicates that simulated SICs are overfitted to observations. Discuss this point.

Response:

We thank the reviewer's comment. In winter, the normalized SIC errors in the control run (averaged over the sea ice-covered regions) are also small (~0.5), indicating that the model simulated SIC matches the satellite measurements well. Significant SIC errors are mainly along the sea ice margins, while the errors in the central Arctic Ocean are usually smaller than the observational errors. We added "indicating that the control run and the satellite SIC measurements match well." In L247-248, we explain why normalized SIC errors are smaller than 0.5.

15. Figure 4, caption: Describe the averaging period for (a)–(c).
Response: We added "**averaged over 2012**" and "**averaged over the sea ice-covered regions**" in the Figure 4 caption.

16. Section 3.2.2: There are no description of Figure 5.
Response:
We thank the reviewer for pointing out this mistake, and the description of Figure 5 was lost during our editing processes. We have included the description of Figure 5 in L285-290.

17. Figure 5, caption: Describe the averaging period.

Response:

We thank the reviewer's comment. The comparisons in Figure 5 are against three mooring-based up-looking-sonar observations covering 2012. To clarify, we have added "2012" on the label of Figure 5 and state it explicitly "throughout the year 2012" in L281.

18. Figures 3, 4, 5, and 6: It might be better to use the same colors among these figures for CTL, adjoint-FD, and adjoint-VP.

Response: We thank the reviewer's advice, and we have remade figures 3-6 with the same line colors for the three simulations among these figures.

19. Line 337: Explain why April 10 and September 20 are chosen for this analysis.

20. Figure 9, caption: Explicitly mention that (a)–(e) are for the control run.

21. Figure 9: It seems that the red lines in (a), (f), and (k) are the September SIE from the control run, the black lines in (g)–(j) are from the adjoint-FD, and those in (k)–(o) are from adjointVP.

Response:

We thank the reviewer's comment. The choice of "April 10-September 20" is that the period covers the melting season of the Arctic sea ice. The purpose of Figure 9 is to show differences in the sea ice retreat process in the control run and the two assimilation runs. Since the reviewers suggested simplifying Figure 9 or splitting Figure 9 into more figures, we have replaced Figure 9 with their corresponding temporal variations integrated over the model domain and rephrased the descriptions in L427-453, concentrating on the significant different sea ice melting processes from May 20-June 15.

22. Line 369: It sounds strange that the SIC change through ice–albedo feedback is categorized as $F_{oi}$ rather than $F_{ai}$.

Response:

  Ice-albedo feedbacks are as follows: sea ice retreat leads to more open water-----surface albedo is reduced----more absorption of solar radiation by sea water----then, the warm water further melts sea ice from the bottom;

    In the model governing equations and during the melting season, **$F_{ai}$ represents the ice-atmosphere heat flux (radiative and turbulent fluxes) at the ice surface. Over the open fraction of seawater, ocean-atmosphere heat fluxes heat the ocean directly, further melting sea ice from the bottom.** Solar heating through the open water is the primary heat source for warming the sea; therefore, we categorize ice-albedo feedback to $F_{oi}$.

---

## Author Comment (AC3)

**Response to Referee**

**By Guokun Lyu on behalf of all coauthors**

The paper introduces new tangent linear and adjoint model for the Viscous-Plastic parameterization of the sea ice model. The key to this work is the stabilization of the non-linear terms following the paper of Toyoda 2019. The novel contribution of this paper was evaluating of the stabilized adjoint in the framework of an Adjoint (ECCO-like) assimilation over the Arctic domain for the calendar year of 2012. I found the paper relevant to its target audience and is generally well written (see few technical comments in the annotated PDF). However, I found that the paper contains a single (but a key) conclusion that is not substantiated by the presented data (see major points below). I suggest that authors introduce new analysis in the revised paper that addresses my concerns (see specific suggestion in the major points section).

Major concerns:

- Ky finding of this paper is summarized in this citation from the manuscript:"Considering the amplitude of air temperature adjustments, the adjustments of the control variables in adjoint-VP are more reasonable than adjoint-FD, and adjoint-VP seems to project model-data misfits to the control variables more reasonably than adjoint-FD." Unfortunately, presented analysis does not provide evidence or error bars on what is reasonable and what is not. This is especially true, given that the authors are using a very old and outdated atmospheric analysis. I suggest that authors augment their paper by the analysis of the observation-minus-first guess errors for control variables that do have direct observations (e.g. wind speed, atmospheric temperature, ocean temperature from profiles). I understand that these measurements are very sparse over the Arctic. Nonetheless some are still available for analysis.

Response:

We thank the referee's comment and suggestion.

The adjoint method adjusts the control variables to make the model simulation consistent with available observations (within prior uncertainties). Therefore, it is more appropriate that the normalized adjustments are ~1.0. That is why we state "adjoint-VP are more reasonable than adjoint-FD" since adjoint-FD over-adjusted 2-m air temperature.

However, adjoint-VP doesn't ensure smaller errors in the adjusted atmosphere variables than the NCEP-RA1 and adjoint-FD at all times and geographic locations. A potential reason is that our coupled ocean-sea ice system doesn't have a dynamic and thermodynamic atmosphere model. We simplified their uncertainties to diagonal; in

this case, the adjoint method couldn't attribute local model-data misfits to remote atmosphere states through atmosphere dynamic processes and statistical correlations.

Since we don't have enough information about the size and spatiotemporal variability of prior uncertainties, we approximated them by the standard deviation of the non-seasonal signals of the NCEP/NCAR-RA1 (in this study) or differences among different atmosphere reanalyses (as in Nguyen et al., 2021). These approximations seem to overestimate their uncertainties as the normalized adjustments are usually smaller than one (~0.3-0.4), except for adjustments of 2-m air temperature.

In the revised manuscript, we use ERA5-NCEP differences as a reference to examine whether the size of the adjustments is reasonable and whether the adjustments may represent the old NCEP-RA1 and newly developed ERA5 reanalyses. As shown in Figures 8 and 11 in the revised manuscript, the adjustments after optimization are much smaller than the ERA5-NCEP differences (<40% of the differences), except the 2-m air temperature in adjoint-FD (>1.5 times the differences during May and July Figure 8c). It is evident that the adjustments in adjoint-VP are much smaller than inter-model (reanalyses) deviations, but adjustments of 2-m air temperature in adjoint-FD are unrealistic (Figure 8c and Figure 11a). These comparisons support our statement "Considering the amplitude of air temperature adjustments, the adjustments of the control variables in adjoint-VP are more reasonable than adjoint-FD, and adjoint-VP seems to project model-data misfits to the control variables more reasonably than adjoint-FD."

- Authors use an obsolete reanalysis product to drive their simulation. While (in itself) their choice does not invalidate their results. I suggest that authors quantify how their choice might impact their conclusions. For example, can the large errors that they report in air temperature corrections can be attributed to a very old reanalysis product?

Response:

We understand the reviewer's concern. If the adjustments represent the differences between new and old version reanalyses, the conclusion that adjoint-VP projects model-data misfits better to the control variables than adjoint-FD will be reversed.

To answer this question, we have compared the differences between the newly developed ERA5 reanalysis and the old NCEP-RA1 reanalysis with the adjustments in adjoint-FD and adjoint-VP (Figures 8, 11). The figures clearly show that the adjustments are much smaller than the ERA5-NCEP differences and don't represent the spatiotemporal patterns of ERA5-NCEP differences. Therefore, we believe that replacing the NCEP-RA1 reanalysis with an updated atmosphere reanalysis will not change the conclusion "adjoint-VP projects model-data misfits to the control variables more reasonably than adjoint-FD" because adjoint-VP represents sea ice process better than adjoint-FD in the adjoint model.

Minor concearns:

- I have attempted to hioghlight a few typos and rough sentences that authors might choose to improve in the revision (see annotated PDF).
- I find that some of the authors figures are very dense and could use more on-figure annotations (e.g. better panel labels). When appropriate, I provide such suggestions in the annotated pdf.

Response:

We thank the reviewer's comments and suggestions. We have revised the typos, and polished the rough sentences. Besides, we remade the busy figures (e.g., Figure 9, Figure 10) and made the explanations and statements more readable.

Nguyen, A. T., Pillar, H., Ocaña, V., Bigdeli, A., Smith, T. A., and Heimbach, P.: The Arctic Subpolar Gyre sTate Estimate: Description and Assessment of a Data-Constrained, Dynamically Consistent Ocean-Sea Ice Estimate for 2002–2017, Journal of Advances in Modeling Earth Systems, 13, e2020MS002398, https://doi.org/10.1029/2020MS002398, 2021.

---

## Referee Report (RR1)

Following is my review of the revised manuscript entitled "Effects of including the adjoint sea ice rheology on estimating Arctic ocean–sea ice state" by Guokun Lyu, Armin Koehl, Xinrong Wu, Meng Zhou, and Detlef Stammer (egusphere-2022-1099).

**General Comment**

In this study, motivated by Toyoda *et al.* (2019), the adjoint sea-ice model with viscous–plastic rheology (adjoint-VP) is applied to a coupled ocean and sea-ice state estimation system for the Arctic Ocean, and compared with the previous version in which the simplified adjoint sea-ice model of free drift (adjoint-FD) is used to avoid numerical instability. One year of optimization experiment for 2012 shows that the adjoint-VP can produce better state of the ocean and sea-ice through more appropriate dynamic and thermodynamic processes than the adjoint-FD.

The revised manuscript became much understandable than the original one in many aspects, but still needs to explain or respond adequately to the following points to be accepted for the publication in *Ocean Science*.

**Specific Comments**

1. Introduction: Indeed, the adjoint method has a characteristic that optimized fields strictly obey the model governing equations, but the control variables are subject to bad influences in some cases as shown in this study (unrealistic adjustment of 2-m air temperature when using the adjoint-FD). In addition, statistical methods can estimate atmospheric forcing and model parameters as well as the initial conditions by augmenting the state vector. Therefore, the reviewer recommends not to exaggerate the advantages of the adjoint method over statistical methods.

2. Line 182: Clarify whether the diffusivity of 500 $m^2 s^{-1}$ is for the vertical or the horizontal.

3. Line 182: Is a harmonic viscosity used in the adjoint model in spite that a biharmonic viscosity is used in the forward model (line 101)?

4. Section 3.2.1: As the authors mention that "the normalized RMSEs in Figure 3d should be close to 1.0 if the optimization found a model simulation consistent with the observations and the prior uncertainties" (line 243), the normalized SIC errors of about 0.5 indicates that simulated SICs are overfitted to observations or the prior uncertainties are overestimated. The same can be said of the normalized RMSEs of SIT (Figure 4). Again, discuss this point.

5. Section 4: Briefly describe the differences between the ERA5 and NCEP-RA1 reanalyses, especially from the viewpoint of the treatment of sea-ice boundary conditions. The reviewer remembers that the NCEP-RA1 does not use a fractional sea-ice concentration but 0 or 1.

6. Section 4: Similar to the specific comment 4, the normalized RMS of adjustments of the atmospheric variables of around 0.1 indicates that their estimated prior uncertainties are too large, or equivalently, the relative contribution of the last term in Equation (1) is too small.

Discuss this point.

7.  Figure 9: It is confusing to use the blue line for adjoint-FD and the black line for adjoint-VP, because they are opposite in other figures.

8.  Figure 11, caption: Explicitly describe the contour intervals.

**Technical Corrections**

1.  The reviewer pointed out the followings in the previous report, but they are not corrected in the revised manuscript.

    1.1  Lines 67: Dynamic should read dynamics.

    1.2  Line 142: 0.25% should read > 25%.

    1.3  Line 170: $C^*$ should be 20.0 rather than –20.0.

    1.4  Line 187 and 192: Dynamic should read dynamics.

    1.5  Line 231: Visual should read visible.

    1.6  Line 470 and 473: Dynamic should read dynamics.

    1.7  Figures 1, 3, 4, 7, and 11: Paint the Great Britain Island gray.

2   Line 101: $m^{-2}\,s^{-1}$ should read $m^2\,s^{-1}$.

3   Line 114: Right hand should read right hand side.

4   Line 447: –6ºC should read 6ºC.

5   Line 449: Figure 10b should read Figure 11b.

---

## Author Response (AR2)

Following is my review of the revised manuscript entitled "Effects of including the adjoint sea ice rheology on estimating Arctic ocean–sea ice state" by Guokun Lyu, Armin Koehl, Xinrong Wu, Meng Zhou, and Detlef Stammer (egusphere-2022-1099).

**General Comment**

In this study, motivated by Toyoda *et al.* (2019), the adjoint sea-ice model with viscous–plastic rheology (adjoint-VP) is applied to a coupled ocean and sea-ice state estimation system for the Arctic Ocean, and compared with the previous version in which the simplified adjoint sea-ice model of free drift (adjoint-FD) is used to avoid numerical instability. One year of optimization experiment for 2012 shows that the adjoint-VP can produce better state of the ocean and seaice through more appropriate dynamic and thermodynamic processes than the adjoint-FD.

The revised manuscript became much understandable than the original one in many aspects, but still needs to explain or respond adequately to the following points to be accepted for the publication in *Ocean Science*.

Response:

We thank the reviewer for carefully reading the manuscript again and apologize for missing the "Technical Corrections" in the last review report.

Based on the reviewer's comments below and in the last review report, we have revised the manuscript and response the reviewer's comment below.

**Specific Comments**

1. Introduction: Indeed, the adjoint method has a characteristic that optimized fields strictly obey the model governing equations, but the control variables are subject to bad influences in some cases as shown in this study (unrealistic adjustment of 2-m air temperature when using the adjoint-FD). In addition, statistical methods can estimate atmospheric forcing and model parameters as well as the initial conditions by augmenting the state vector. Therefore, the reviewer recommends not to exaggerate the advantages of the adjoint method over statistical methods.

Response:

We thank the reviewer's suggestion. The main propose of such a comparison is to explain advantage/disadvantage of different methods.

To achieve a more complete and fair comparisons between statistical and adjoint methods, we revised the manuscript in the two aspects:1) we remove the improper statement "In addition, the adjoint method adjusts all uncertain inputs, including initial conditions, atmospheric forcing, and model parameters, rather than only the initial conditions as in statistical-based methods."; 2) we point out the disadvantage of adjoint method "However, the qualities the reanalysis dataset depends on the accuracy of the tangent linear approximation."

In this way, we hope the readers could have better and fair understandings on different data assimilation methods.

2. Line 182: Clarify whether the diffusivity of 500 $m^2\,s^{-1}$ is for the vertical or the horizontal.

Response: 500 $m^2\,s^{-1}$ is horizontal diffusivity. We have clarified it in the manuscript.

3. Line 182: Is a harmonic viscosity used in the adjoint model in spite that a biharmonic viscosity is used in the forward model (line 101)?

Response:

We thank the reviewer's comment. Here we explain why we "**increase (or add)**" harmonic viscosity in the adjoint model, while in the forward model we use biharmonic viscosity.

1) In the adjoint model, the biharmonic viscosity is included with the same coefficient as in the forward model.

2) We further add harmonic viscosity and tracer diffusion to stabilize the adjoint model over a large assimilation window. We explain the reason as follows.

Assuming the forward model is expressed as:

$$y = Mx$$

Where x is model input, y is model state, and M represents model integration matrix. Its tangent linear approximation is

$$\Delta y = M'(x)\Delta x.$$

M'(x) is tangent linear model, $\Delta x$ is perturbations of model inputs and $\Delta y$ is the resulting model state perturbations. For chaotic nonlinear system, such as the coupled ocean-sea ice system in this study, the nonlinear system (M) and its tangent linear system (M'(x)) have positive Lyapunov exponents and eigenvalues, respectively, limiting the assimilation window. We need to add extra terms in the tangent linear/adjoint systems to reduce/remove the positive eigenvalues in the adjoint model to extend the assimilation window. In the model implementation, both harmonic and biharmonic viscosity and diffusion can be used to damp the positive eigenvalues in the adjoint model. We choose to use harmonic viscosity because it is a more efficient to damping out fast growing modes than the biharmonic ones.

4. Section 3.2.1: As the authors mention that "the normalized RMSEs in Figure 3d should be close to 1.0 if the optimization found a model simulation consistent with the observations and the prior uncertainties" (line 243), the normalized SIC errors of about 0.5 indicates that simulated SICs are overfitted to observations or the prior uncertainties are overestimated. The same can be said of the normalized RMSEs of SIT (Figure 4). Again, discuss this point.

Response:

We thank the reviewer's comment. We have added discussion on the too large SIT uncertainties and pose the requirements on more accurate SIT observations in the future.

5. Section 4: Briefly describe the differences between the ERA5 and NCEP-RA1 reanalyses, especially from the viewpoint of the treatment of sea-ice boundary conditions. The reviewer remembers that the NCEP-RA1 does not use a fractional sea-ice concentration but 0 or 1.

Response:

We thank the reviewer's comment. We have added more comments on differences between the NCEP-RA1 and ERA5 lower boundary conditions "The ERA5 uses

fractional SIC as surface boundary conditions, but NCEP-RA1 uses 0 and 1 for ice-free and ice-covered ocean, respectively." (L359-L361).

6. Section 4: Similar to the specific comment 4, the normalized RMS of adjustments of the atmospheric variables of around 0.1 indicates that their estimated prior uncertainties are too large, or equivalently, the relative contribution of the last term in Equation (1) is too small. Discuss this point.
Response:
We thank the reviewer's comment. By now, the uncertainties of atmosphere states remain uncertain. We use the standard deviations of the non-seasonal signals from NCEP-RA1. Nguyen et al. (2021) base on differences/deviations of different atmosphere reanalysis. But, both work show that the adjustments are much smaller than the estimated prior uncertainties. We have added comments on the small values of normalized adjustments "The normalized adjustments of 0.1-0.6 indicate that the estimated prior uncertainties of atmospheric state remain too large." L357.

7. Figure 9: It is confusing to use the blue line for adjoint-FD and the black line for adjoint-VP, because they are opposite in other figures.
Response: we thank the reviewer's comment. We have revised Figure 9 to make the line colors/experiments the same as in the other Figures.

8. Figure 11, caption: Explicitly describe the contour intervals.
Response: we have added "The contour intervals are 2 °C" in Figure 11 caption.

**Technical Corrections**
1. The reviewer pointed out the followings in the previous report, but they are not corrected in the revised manuscript.
Response:
    We thank the reviewer for carefully reading throughout the manuscript and apologize for missing the "technical corrections" in the previous review report.
    Based on the previous report (technical corrections) and manuscript, corrections below and revised manuscript last time, we have revised the manuscript.

1.1 Lines 67: Dynamic should read dynamics.
Response: we have corrected the mistakes throughout the paper.
1.2 Line 142: 0.25% should read > 25%.
Response: we thank the reviewer and we have revised the mistake.
1.3 Line 170: $C^*$ should be 20.0 rather than –20.0.
Response: we have corrected this mistake.

1.4 Line 187 and 192: Dynamic should read dynamics.
Response: we thank the reviewer. We have corrected the mistakes throughout the manuscript.

1.5 Line 231: Visual should read visible.

Response: we have change "visual" to visible in the context.

1.6 Line 470 and 473: Dynamic should read dynamics.

Response: we thank the reviewer. We have corrected the mistakes throughout the manuscript.

1.7 Figures 1, 3, 4, 7, and 11: Paint the Great Britain Island gray.

Response:

We thank the reviewer for pointing out the mistakes. We have replotted Figures 1,3,4,7,11 and painted Great Britain Island gray.

2 Line 101: $m^{-2} s^{-1}$ should read $m^2 s^{-1}$.

Response: we have revised the units $m^{-2} s^{-1}$ to $m^2 s^{-1}$

3 Line 114: Right hand should read right hand side.

Response: We have changed "right hand" to "right hand side".

4 Line 447: –6ºC should read 6ºC.

Response: we have removed "–"

5 Line 449: Figure 10b should read Figure 11b

Response: we thank the reviewer for pointing out the mistakes. We have changed "Figure 10b" to "Figure 11b" here.

Nguyen, A. T., Pillar, H., Ocaña, V., Bigdeli, A., Smith, T. A., & Heimbach, P. (2021). The Arctic Subpolar Gyre sTate Estimate: Description and Assessment of a Data-Constrained, Dynamically Consistent Ocean-Sea Ice Estimate for 2002–2017. *Journal of Advances in Modeling Earth Systems*, *13*(5), e2020MS002398. https://doi.org/https://doi.org/10.1029/2020MS002398